# Guaranteed efficient energy estimation of quantum many-body Hamiltonians using ShadowGrouping

Alexander Gresch [1,2] ✉ & Martin Kliesch [2] ✉

Estimation of the energy of quantum many-body systems is a paradigmatic task in various research fields. In particular, efficient energy estimation may be crucial in achieving a quantum advantage for a practically relevant problem. For instance, the measurement effort poses a critical bottleneck for variational quantum algorithms. We aim to find the optimal strategy with single-qubit measurements that yields the highest provable accuracy given a total measurement budget. As a central tool, we establish tail bounds for empirical estimators of the energy. They are helpful for identifying measurement settings that improve the energy estimate the most. This task constitutes an **NP**-hard problem. However, we are able to circumvent this bottleneck and use the tail bounds to develop a practical, efficient estimation strategy, which we call ShadowGrouping. As the name indicates, it combines shadow estimation methods with grouping strategies for Pauli strings. In numerical experiments, we demonstrate that ShadowGrouping improves upon state-of-the-art methods in estimating the electronic ground-state energies of various small molecules, both in provable and practical accuracy benchmarks. Hence, this work provides a promising way, e.g., to tackle the measurement bottleneck associated with quantum many-body Hamiltonians.

As their name suggests, observables are said to be the physically observable quantities in quantum mechanics. Their expectation values play a paradigmatic role in quantum physics. However, quantum measurements are probabilistic, and, in practice, expectation values have to be estimated from many samples, i.e., many repetitions of experiments. The arguably most important observables, such as quantum many-body Hamiltonians, cannot be measured directly but have some natural decomposition into local terms. Typically, they are estimated individually, in commuting groups[1–6], or using randomized measurements[7–12] to keep the number of samples sufficiently low. So far, the focus has been on estimating the local terms first with individual error control and then combining them into the final estimate. Sample complexity bounds fully tailored to the estimation of many-body Hamiltonians are still missing.

Energy estimation from not too many samples is becoming an increasingly critical task in applications. After advances on quantum supremacy[13,14], achieving a practical quantum advantage has now arguably become the main goal in our field. The perhaps most promising practical application is the simulation of physical systems[15], as already suggested by Feynman[16]. The estimation of ground states of quantum many-body Hamiltonians plays a paradigmatic role in this endeavor. The two main ways to solve this task are (i) a digital readout of the energy as achieved by the phase estimation algorithm and (ii) a direct readout. Since (i) seems to require fault-tolerant quantum computation, which is out of reach at the moment, we focus on (ii) with particularly simple direct readout strategies that seem most amenable to noisy and intermediate scale quantum (NISQ)[17] hardware.

As one concrete possible application of our energy estimation strategy, we discuss VQAs. In VQAs, one aims to only use short

[1]Faculty of Mathematics and Natural Sciences, Heinrich Heine University Düsseldorf, Düsseldorf, Germany. [2]Institute for Quantum Inspired and Quantum Optimization, Hamburg University of Technology, Hamburg, Germany. ✉e-mail: alexander.gresch@hhu.de; martin.kliesch@tuhh.de

parametrized quantum circuit (PQCs) in order to finish the computation before the inevitable noise has accumulated too much. The most promising, yet challenging computational problems come from quantum chemistry or combinatorial optimization for which the variational quantum eigensolver[18–20] and the quantum approximate optimization algorithm (QAOA)[21,22] have been proposed, respectively. In either case, we aim to find the ground state of a given Hamiltonian $H$ by preparing a suitable trial state $\rho$ via the PQC. Its parameters need a classical optimization routine, often done via gradient-based methods. In this case, the estimation of the gradient itself can be restated as an energy estimation task by using a parameter-shift rule[23–31]. Therefore, the elementary energy estimation task remains even if the actual ground state lies in the ansatz class of the VQA and if obstacles such as barren plateaus[32] or getting stuck in local minima[33,34] are avoided. We refer to the review articles[35,36] for more details.

Analog quantum simulators are another very promising approach to achieve a useful quantum advantage[37]. In these approaches, a target state $\rho$ associated to a quantum many-body Hamiltonian is prepared, which could be a time-evolved state, a thermal state, or a ground state. Given this preparation, one or more observables of interest have to be measured to infer insights about $\rho$. For instance, they could be some spin or particle densities, correlation functions, or an energy. All these observables are, however, captured by $k$-local observables. There are various possibilities of how such quantum simulations could provide a useful quantum advantage on imperfect hardware[38]. Typically, analog quantum simulators are limited in their readout capabilities. At the same time, the single-qubit control is rapidly improving, rendering them a more and more contesting alternative.

As each measurement requires its own copy of $\rho$, i.e., preparing the state again for each measurement, this constitutes a huge bottleneck. This is especially true in quantum chemistry applications where we require a high precision for the final energy estimate of the (optimized) state. The resulting bottleneck is persistent no matter how we may design the actual PQC preparing the trial state or which quantum simulator is considered. As a consequence, tackling the measurement bottleneck is crucial for any feasible application of NISQ-friendly hardware to any practicable task. Hence, in order to keep the energy estimation feasible, reliable and controlled, we ask for the following list of desiderata to be fulfilled. The energy estimation protocol should be

(i)   based only on basis measurements and single-qubit rotations,

(ii)  it comes along with rigorous guarantees and sample complexity bounds for the energy estimation,

(iii) the required classical computation must be practically feasible, and

(iv)  it should yield competitive results to state-of-the-art approaches.

Previous works addressed these points mostly separately. For such settings, two main paradigms for the energy estimation task have emerged: grouping strategies[1–6] and (biased) classical shadows[7–12,39] as well as a first framework to partially unify the two[40]. We provide some more details in the Supplementary Information. A few ideas outside these paradigms also exist[41–43]. Most of these works are compatible with (i) and (iii) and fulfill (iv). However, the metrics introduced to track the amount of measurement reduction achieved leave (ii) unfulfilled. This lack of guarantees is pernicious for two reasons. On one hand, we want to be able to efficiently estimate Hamiltonian expectation values (or any other Hermitian operator for that matter) in relevant quantum experiments where the actual solution is not known and the qubits' number exceeds those used in the addressable benchmarks. In quantum chemistry applications, for example, high precision is priority and a guarantee for the estimation error is key. On the other hand, obtaining sample complexities for these quantum algorithms is vital in accessing their feasibility in reliably addressing problems with increasing number of qubits. The current benchmarks already hint at a daunting measurement effort despite not even exceeding 20 qubits.

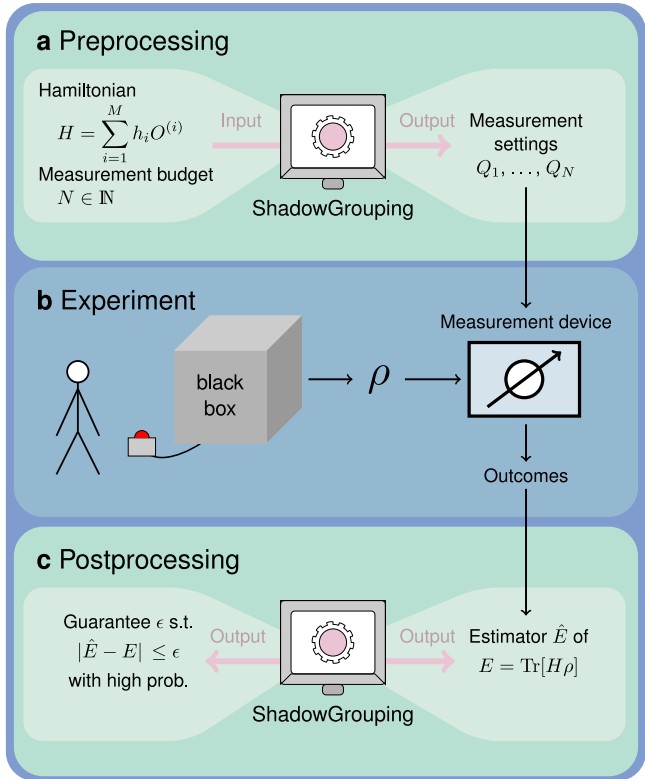

**Fig. 1 | Overview of our estimation protocol. a** As input, we are given a description of the Hamiltonian $H$ in terms of Pauli observables $O^{(i)}$ and a measurement budget $N$. ShadowGrouping generates a list of measurement settings $(Q_i)_{i=1}^N$ in a pre-processing step. **b** Then, $N$ copies of an unknown quantum state $\rho$ are prepared sequentially and measured such that the $i$-th copy is measured in the setting given by $Q_i$. The measurements result in $N$ bit strings as measurement outcomes. **c** In the postprocessing, the description of $H$ and the measurement outcomes are combined into the estimator $\hat{E}$ of the state's energy $E$ together with an accuracy upper bound $\epsilon$. The protocol works independently of the strategy that is used to generate the settings, which automatically results in different estimators and bounds $\epsilon$. Conversely, one can also minimize $\epsilon$ to obtain suitable measurement settings $(Q_i)_{i=1}^N$.

Understanding how the sample complexity of an energy estimation task and a particular choice for the measurement strategy scales with the number of qubits enables the user to forecast their chance of successfully completing the task beforehand.

In this work, we improve upon state-of-the-art estimation protocols and provide rigorous guarantees completing desideratum (ii). We summarize the estimation task and our contributions contained in the Results section in Fig. 1. In particular, they include rigorous guarantees for commonly used grouping techniques. We do so by providing tail bounds on empirical estimators of the state's energy that are compatible with grouping strategies. This way, our bound allows us to assess the accuracy and feasibility of typical state-of-the-art measurement schemes. We show that minimizing this upper bound is **NP**-hard in the number of qubits in the worst case. As a heuristic solution, we propose our own measurement scheme which we call ShadowGrouping that efficiently makes use of the observables' dominating contributions to the upper bound as well as their respective commutation relations. We conclude with an outlook in the Discussion section.

## Results

We structure our results as follows. First, we provide the provable guarantees for measurement strategies. This subsection also includes the hardness of finding optimal measurement settings in the number

of qubits, which shows that heuristic measurement optimization approaches are required. In particular, the hardness result motivates the conception of ShadowGrouping, presented in the subsequent subsection. Finally, we numerically demonstrate that ShadowGrouping improves upon other state-of-the-art approaches in the benchmark of estimating the electronic ground-state energy of various small molecules.

Throughout this work, $[k]$ denotes the set $\{1,\ldots,k\}$ and we set $\mathcal{P} = \{X,Y,Z\}$ as shorthand notation for labels of the Pauli matrices and $\mathcal{P}^n$ for Pauli strings, i.e., labels for tensor products of Pauli matrices. Moreover, let $\mathcal{P}_1 = \{\mathbb{1},X,Y,Z\}$ and $\mathcal{P}_1^n$ be defined analogously. Finally, the $p$-norm of a vector $\boldsymbol{x}$ is denoted as $\|\boldsymbol{x}\|_{\ell_p}$ and the absolute value of any $x \in \mathbb{C}$ as $|x|$.

## Equipping measurement strategies with provable guarantees

In order to set the stage, we properly define the energy estimation task and give a notion of a measurement scheme. Assume we are handed an $n$-qubit quantum state $\rho$ of which we want to determine its energy $E$ w.r.t. a given Hamiltonian $H$. The energy estimation is not a straightforward task: due to the probabilistic nature of quantum mechanics, we have to estimate $E$ by many measurement rounds in which we prepare $\rho$ and measure it in some chosen basis. Moreover, we typically cannot measure the state's energy directly. Instead, we assume the Hamiltonian to be given in terms of the Pauli basis as

$$H = \sum_{i=1}^{M} h_i O^{(i)}, \ O^{(i)} = \bigotimes_{j=1}^{n} O_j^{(i)} \tag{1}$$

with $h_i \in \mathbb{R}$ and single-qubit Pauli operators $O_j^{(i)} \in \mathcal{P}_1$. Often, we identify $H$ with its decomposition

$$H \equiv \left( h_i, O^{(i)} \right)_{i \in [M]}. \tag{2}$$

Without loss of generality, we assume that $O^{(i)} \neq \mathbb{1}^{\otimes n} \forall i$. To ensure the feasibility of this decomposition, we require that $M = \mathrm{O}(\mathrm{poly}(N))$. This is the case, for example, in quantum chemistry applications where $M$ scales as $n^4$.

Given a quantum state $\rho$, the energy estimate is determined by evaluating each expectation value $o^{(i)} := \mathrm{Tr}[\rho \, O^{(i)}]$. By $\hat{o}^{(i)}$ we denote the empirical estimator of $o^{(i)}$ from $N_i$ samples. In more detail, $\hat{o}^{(i)} := \frac{1}{N_i} \sum_{\alpha=1}^{N_i} y_\alpha$, where $y_\alpha \in \{-1, 1\}$ are iid. random variables determined by Born's rule $\mathbb{P}[y_\alpha = 1] = \mathrm{Tr}[\rho \, (Q_i - \mathbb{1})/2]$. We assume that each $\hat{o}^{(i)}$ is estimated from iid. preparations of $\rho$. This assumption solely stems from the proof techniques of the classical shadows[10] used in order to arrive at our estimator (3) below. We expect this assumption to be loosened in the future such that we only need to assume unbiased estimators $\hat{o}^{(i)}$. In either case, we do not assume different $\hat{o}^{(i)}$ to be independent. In particular, we can reuse the same sample to yield estimates for multiple, pair-wise commuting observables at once.

Leveraging standard commutation relations requires many two-qubit gates for the readout, increasing the noise in the experiment or quantum circuit enormously. Therefore, we impose the stronger condition of qubit-wise commutativity (QWC): any two Pauli strings $P = \bigotimes_i P_i, Q = \bigotimes_i Q_i$ commute qubit-wise if $P_i$ and $Q_i$ commute for all $i \in [n]$. Again, the empirical estimators $\hat{o}^{(i)}$ do not have to be independent as a consequence of using the same samples for the estimation of several (qubit-wise) commuting observables. Using these estimators, the energy can be determined. By linearity of Eq. (2) we have that

$$E = \sum_{i=1}^{M} h_i o^{(i)}, \ \hat{E} = \sum_{i=1}^{M} h_i \hat{o}^{(i)} \tag{3}$$

to which we refer as the grouped empirical mean estimator.

For conciseness, we introduce our notions of measurement settings, schemes and compatible Pauli strings in Definitions 1 and 2 in the following.

**Definition 1.** Let $H$ be a Hamiltonian of the form (2) and $N \in \mathbb{N}$ a number of measurement shots. An algorithm $\mathcal{A}$ is called a measurement scheme if it takes $(H, N)$ as input and returns a list of measurement settings $\boldsymbol{Q} \in (\mathcal{P}^n)^N$ specifying a setting for each measurement shot.

Having formalized what a measurement schemes does, we have to take a look at the Pauli strings in the Hamiltonian decomposition (2) and their commutation relations as they effectively require various distinct measurement settings to yield an unbiased estimate of the energy. To this end, we define how we can relate the target Pauli strings to a proposed measurement setting by means of compatible measurements:

**Definition 2.** Consider a Pauli string $Q \in \mathcal{P}^n$ as a measurement setting. A Pauli string $O \in \mathcal{P}_1^n$ is said to be compatible with $Q$ if $O$ and $Q$ commute. Furthermore, they are QWC-compatible if $O$ and $Q$ commute qubit-wise. We define the compatibility indicator $\mathrm{C} : \mathcal{P}_1^n \times \mathcal{P}_1^n \to \{\mathrm{True} \equiv 1, \mathrm{False} \equiv 0\}$ such that $\mathrm{C}[O,Q] = \mathrm{True}$ if and only if $O$ and $Q$ are compatible. Analogously, we define $\mathrm{C}_{\mathrm{QWC}}$ that indicates QWC-compatibility.

With these two definitions, we are able to formalize what we mean by equipping a measurement scheme with guarantees. As sketched in Fig. 1, we are given access to a device or experiment that prepares an unknown quantum state $\rho$ and some Hamiltonian (2). Not only do we want to estimate its energy from repeated measurements, but we would like to accommodate the energy estimator with rigorous tail bounds. That is, we wish to determine how close the estimate $\hat{E}$ is to the actual but unknown energy $E$ and how confident we can be about this closeness. Mathematically, we capture the two questions by the failure probability, i.e., the probability that $|\hat{E} - E| \geq \epsilon$ for a given estimation error $\epsilon > 0$. In general, this quantity cannot be efficiently evaluated (as it depends on the unknown quantum state produced in the experiment). Nevertheless, we can often provide upper bounds to it that hold regardless of the quantum state under consideration. One crucial requirement is that we simultaneously want to minimize the total number of measurement rounds. For instance, in grouping strategies we extract multiple samples from a single measurement outcome. Here, the grouping is carried out such that the variance, $\mathrm{Var}[\hat{E}]$, of the unbiased estimator (3) is minimized. By virtue of Chebyshev's inequality, this serves as an upper bound to the failure probability. However, $\mathrm{Var}[\hat{E}]$ is neither known and one has to rely on estimating it by empirical (co)variances of the Pauli observables. This, in turn, introduces an additional error that is both unaccounted for and in general not negligible for finite measurement budgets[44][Proposition 4]. However, due to the introduced correlation between samples for commuting Pauli terms in the Hamiltonian (2), standard arguments based on basic tail bounds cannot be applied. We resolve both issues by formulating a modified version of the vector Bernstein inequality[45,46] in a first step to bound the joint estimation error of each of the contributing Pauli observables. In particular, this takes into account any correlated samples that stem from the same measurement round. In the same step, we extend the inequality to arbitrary random variables in a separable Banach space which might be of independent interest. Afterwards, we show that this actually serves as an upper bound of the absolute error of the energy estimation. In particular within the grouping framework, this signifies a paradigm shift: our guarantees are a step towards fulfilling desideratum (ii) unconditionally, i.e., without having to rely on estimating the variance of the estimator in the first place.

The energy estimation inconfidence bound reads as follows.

**Theorem 3.** Consider $\boldsymbol{Q}$ obtained from a measurement scheme for some input $(H, N)$. Let $\delta \in (0, 1/2)$. Fix a compatibility indicator $\mathrm{f} = \mathrm{C}$ or

$f = C_{QWC}$. Denote the number of compatible measurements for observable $O^{(i)}$ by $N_i(\boldsymbol{Q}) := \sum_{j=1}^{N} f(Q_j, O^{(i)})$ and assume $N_i \geq 1$ for all $i \in [M]$ (we usually drop the $\boldsymbol{Q}$-dependence). Denote $h_i' := |h_i|/\sqrt{N_i}$ and $h_i'' := |h_i|/N_i$. Moreover, let $2\|\boldsymbol{h}'\|_{\ell_1} \leq \epsilon \leq 2\|\boldsymbol{h}'\|_{\ell_1}(1 + \|\boldsymbol{h}'\|_{\ell_1}/\|\boldsymbol{h}''\|_{\ell_1})$. Then any grouped empirical mean estimator (3) satisfies

$$\mathbb{P}\left[|\hat{E} - E| \geq \epsilon\right] \leq \exp\left(-\frac{1}{4}\left[\frac{\epsilon}{2\|\boldsymbol{h}'\|_{\ell_1}} - 1\right]^2\right). \quad (4)$$

We sketch a proof of this theorem in the Methods section and provide a detailed proof in the Supplementary Information. This result shows that we can equip any measurement scheme with guarantees, which hold uniformly for all quantum states. In particular, it is compatible with correlated samples, rendering it applicable to popular grouping strategies. Additionally, Theorem 3 also serves as a benchmarking tool: given a Hamiltonian decomposition (2) and a confidence $\delta \in (0, 1/2)$, we can compare any two measurement strategies, each of them preparing a certain number of measurement settings: We set the right-hand side of Eq. (4) equal to $\delta$ and solve for $\epsilon$. This calculation yields the error bound

$$\epsilon \leq 6 \log \frac{1}{\delta} \|\boldsymbol{h}'\|_{\ell_1}, \quad (5)$$

which is automatically larger than $2\|\boldsymbol{h}'\|_{\ell_1}$. It has a similar scaling as the expected statistical deviation in ref. 6 [Eq. (13)] but is stronger in the sense that it is a tail-bound-based guarantee. The minimization of $\epsilon$ over $\boldsymbol{Q}$ we refer to as the optimization of a measurement scheme.

One option for this optimization is to introduce a small systematic error in favor of a larger statistical error, i.e., to introduce a biased estimator of the energy. A straightforward idea is to remove certain observables from the Hamiltonian, i.e., a truncation of the Pauli decomposition[20]. In Corollary 12 in the Supplementary Information, we find conditions (based on Theorem 3) under which such a truncation is justified. Interestingly, this does not depend on the coefficients of the Pauli terms but only on how many compatible measurement settings we have available. Another idea to increase the guaranteed precision is to optimize over the measurement setting $\boldsymbol{Q}$. However, this optimization is NP-hard in the number of qubits:

**Proposition 4.** Consider a Hamiltonian (2), state $\rho$, the grouped empirical mean estimator (3) and $N \geq 1$ a number of measurement settings. Then, finding the measurements settings $\boldsymbol{Q} \in (\mathcal{P}^n)^N$ that minimize a reasonable upper bound to $\mathbb{P}[|\hat{E} - E| \geq \epsilon]$, such as Eq. (4), is NP-hard in the number of qubits $n$. In particular, it is even NP-hard to find a single measurement setting that lowers this bound the most.

The formal statement of Proposition 4 and its proof are contained in the Supplementary Information. In summary, we show the hardness by reducing the optimization of the measurement scheme from a commonly used grouping technique. Since finding the optimal grouping strategy is known to be NP-hard[40,47], this also transfers over to the optimization of the measurement scheme. Therefore, we have to rely on heuristic approaches to practically find suitable measurement settings. In the following, we devise our own efficient measurement scheme that is aware of both the upper bound and the commutation relations among the Pauli observables to find such settings.

## ShadowGrouping
We aim to determine the energy $E$ by measuring the individual Pauli observables in Eq. (2). In order to increase the accuracy of the prediction with the smallest number of measurement shots possible, Theorem 3 suggests minimizing Eq. (5). The minimization is done by choosing the most informative measurement settings by exploiting

the commutativity relations of the terms in the Hamiltonian decomposition. However, Proposition 4 states that finding the next measurement settings that reduce the current inconfidence bound the most is NP-hard, even when trying to find a single measurement setting. As a suitable heuristic, we propose an approach that makes use of the structure of the terms in the tail bound and which we call ShadowGrouping. It makes use of the fact that there exists a natural hierarchy for each of the terms in the decomposition: we order the Pauli observables decreasingly by their respective importance to the current inconfidence bound. This gives rise to a non-negative weight function weight that takes the Hamiltonian decomposition (2) and a list of previous measurement settings $\boldsymbol{Q}$ as inputs and outputs a non-negative weight $w_i$ for each Pauli observable in the decomposition. Here, the weight is defined as

$$\text{weight}(\boldsymbol{Q}, H)_i := |h_i| \frac{\sqrt{N_i + 1} - \sqrt{N_i}}{\sqrt{N_i(N_i + 1)}} > 0 \quad (6)$$
$$\text{with } N_i = N_i(\boldsymbol{Q}).$$

Details on the motivation for this choice can be found in the Methods section. The function weight takes into account two key properties: the importance $|h_i|$ of each observable in the Hamiltonian (2) and how many compatible measurement settings we collected previously. A larger weight increases the corresponding observable's contribution to the bound as statistical uncertainties get magnified. On the other hand, this uncertainty is decreased by collecting more compatible settings. As the weights are derived from tail bounds to the estimation error, the individual contributions dwindle when increasing the number of compatible measurement settings. Iterative application of ShadowGrouping thus ensures that each observable eventually has at least one compatible measurement setting.

We now explain how ShadowGrouping utilizes these weights to find measurement settings tailored to the Hamiltonian. A sketch of the algorithm is presented in Fig. 2 but we also provide the pseudocode for ShadowGrouping in Box 1. This routine works for both QWC and general commutativity, respectively. We call the idle part of a measurement setting $Q$ the set of those qubits where $Q$ acts as the identity. The idea of the algorithm is as follows. We start with an idle measurement setting $Q = \mathbb{1}^{\otimes n}$. First, we order the observable list by their respective value of weight. Assume $O$ to be the next element from this ordered list. For the QWC-case, we simply check whether $C_{QWC}[O, Q]$, see Definition 2. If so, we allow changing the idle parts of $Q$ to match $O$.

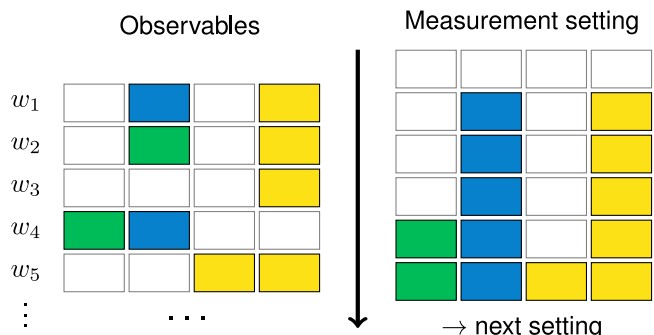

**Fig. 2 | Sketch of the generation of a measurement setting $\boldsymbol{Q} \in \mathcal{P}^n$.** Each box corresponds to a single-qubit operator in the tensor product. Empty boxes correspond to $\mathbb{1}$ and the three colors to each of the Pauli operators, respectively. We order the observables descendingly by their respective weights, i.e., $w_1 \geq w_2 \geq w_3 \geq \ldots \geq w_M$, each of which is computed with the function weight. The arrow in the middle indicates the order in which $Q$ is adjusted to the observables indicated on the left. Only the observable in the second row cannot be measured with the measurement setting proposed by the algorithm. After single-qubit measurements have been assigned, they cannot be altered anymore to ensure compatibility with previously considered observables.

## BOX 1
# ShadowGrouping algorithm

**Require** Hamiltonian decomposition $H = (h_i, O^{(i)})_{i \in [M]}$

**Require** previous measurement settings $\mathbf{Q} \in (\mathcal{P}^n)^{N-1}$

**Require** function weight() to attribute a weight to each observable, e.g., Eq. (6)

**Require** boolean function *CHECK_EVEN*() that counts the number of qubits on which two Pauli strings do not have QWC and returns whether it is even

**Require** compatibility indicator f = C or $C_{QWC}$

1: $Q \leftarrow \mathbb{1}^{\otimes n}$
2: $w_i \leftarrow \text{weight}(\mathbf{Q}, H)_i \; \forall i$
3: **While** $|\text{supp}(Q)| < n$ **and** $\max_i w_i > 0$ **do**
4: $j \leftarrow \arg\max_i w_i$ ▷ or use *ARGSORT*() instead
5: $S \leftarrow \text{supp}(O^{(j)}) \backslash \text{supp}(Q)$ ▷ relevant idle parts in Q
6: **if** f = C **and not** *CHECK_EVEN*($O^{(j)}, Q$) **then**
7: pick $i \in S$ at random
8: $Q_i \leftarrow \mathcal{P} \backslash \{O_i^{(j)}\}$ at random
9: $S \leftarrow S \backslash \{i\}$
10: **end if**
11: **if** f[$O^{(j)}, Q$]
12: update Q s.t. $O_i^{(j)} = Q_i \; \forall i \in \mathcal{S}$
13: **end if**
14: $w_j \leftarrow 0$
15: **end while**
16: **return** Q

For example, $C_{QWC}[X\mathbb{1}, \mathbb{1}Y]$, thus we would alter $\mathbb{1}Y$ into $XY$ in Line 12 of Box 1. The case of general commutativity is slightly more involved and we have to inspect three cases. If the support of the observable $O$ to be checked is disjoint with the support of $Q$, $O$ and $Q$ commute (and we update as above). Updating $Q$'s remaining idle parts at later stages of the algorithm does not change that. If the support of $O$, on the other hand, is a subset of the support of $Q$, we only need to check the commutativity on $O$'s support and never update $Q$ regardless of the outcome. If neither case, we first check whether $O$ and $Q$ commute when restricting to the support of $Q$. If not, we know that $O$ and $Q$ break QWC on an odd number of qubits[1]. For example, observable $O = YY$ and $Q = \mathbb{1}X$ break QWC on a single qubit, but $YY$ does commute with both $ZX$ and $XX$. Thus, as long as there are idle parts in $Q$ left, we can alter $Q$ at a single qubit such that it commutes with $O$. Eventually, there are no more identities in $Q$ left or all observables have been checked. In either case, we have found the next measurement setting. If we consider QWC, each element in $Q$ directly tells us in which Pauli basis the corresponding qubit has to be measured. If general commutativity is to be considered, we can simply run through the list again and keep track of all observables commuting with $Q$. From this set, we can construct a suitable quantum circuit to measure them jointly[4]. In either case, we refresh the weights (6) afterwards according to the updated $N_i$, and iterate.

Our algorithm has two major advantages over state-of-the-art strategies. First, our algorithm is highly adaptable: it only requires a weight function weight that provides a hierachy for the Pauli observables. Moreover, in case weight is derived from an actual upper bound like Eq. (4), we can adapt the hierarchy after each round while improving the guarantees from Theorem 3. This does not require carrying out the readout, we merely keep track of previous settings. This way, we can apply ShadowGrouping in an on-line setting as we do not require a costly preprocessing step as in typical grouping

schemes[6,40]. As another consequence, our scheme is capable of adapting to previous measurement settings, similar to the Derandomization approach of ref. 10. Secondly, the algorithm is also efficient: each pass through the algorithm has a computational complexity of $O(M \log(M))$ due to the sorting of the $M$ weighted observables. Updating the measurement setting $Q$ while traversing the ordered list ensures that the complexity does not increase further. Standard grouping techniques, on the other hand, compute the whole commutativity graph which requires $O(M^2)$. Our procedure thus corresponds to a continuously adapting overlapped grouping strategy[40] but also incorporates the performance guarantees obtained from the classical shadow paradigm, hence our naming scheme.

## Numerical benchmark
One common benchmark to compare the performance of the various measurement schemes is the estimation of the electronic ground-state energy $E$ of various small molecules[10]. The fermionic Hamiltonian given a molecular basis set has been obtained using Qiskit[48] which also provides three standard fermion-to-qubit mappings: Jordan-Wigner (JW)[49], Bravyi-Kitaev (BK)[50] and the Parity transformation[50,51]. Then, the Hamiltonian is exact diagonalized to obtain the state vector of the ground state and its energy $E$ for the benchmark. Together, we obtain the Hamiltonian decomposition (2) and are able to run the various measurement schemes to obtain an estimate $\hat{E}$ of $E$ by repeatedly drawing samples from the state. The code generating the results can be found in a separate repository[52]. We compare ShadowGrouping with other state-of-the-art methods such as Overlapped Grouping (with parameter $T = 10^7$)[40], Adaptive Pauli estimation[11], Derandomization (with default hyperparameter $\epsilon^2 = 0.9$[10] and AEQuO (with parameters $L = 3$, $l = 4$ and $N_{\text{tot}} = 10^5$)[6]. To keep the comparison fair, we only examine methods that utilize Pauli basis measurements without any additional two-qubit gates. This excludes grouping methods that focus solely on general commutation relations[1–5].

For the benchmark, we consecutively generate a single measurement setting from a given measurement scheme and measure it once using qibo[53]. We repeat this procedure $N$ times to yield an energy estimate $\hat{E}_N$ over $N$ measurement outcomes. Increasing $N$ successively, we keep track of two measures for the estimation quality. The first one we refer to as the empirical measure since we choose the root mean square error (RMSE) defined as

$$\text{RMSE}_N := \sqrt{\frac{1}{N_{\text{runs}}} \sum_{i=1}^{N_{\text{runs}}} \left( \hat{E}_N^{(i)} - E \right)^2} \tag{7}$$

over $N_{\text{runs}} = 100$ independent runs. This measure requires knowledge of the solution to the problem to be solved and can, hence, only be applied to problem instances of small system sizes. Nevertheless, its direct correspondence to the quantity of interest, the estimation error $|\hat{E} - E|$, renders it a suitable benchmarking tool on known cases.

For general cases, however, we can investigate our guarantee instead, which does not require the knowledge of any ground-state energy. By virtue of Theorem 3, we have access to guarantees for the estimation accuracy given the proposed measurement settings. Using only these settings of each measurement scheme, i.e., without preparing the quantum state $\rho$, we calculate the corresponding guaranteed accuracy. Because we do not require any state dependence (as no samples are drawn from $\rho$ at all), this comparison yields a rigorous and practical comparison for the schemes. With this approachable figure of merit, we again generate measurement settings and track the guaranteed accuracy $\epsilon_{\mathcal{A}}$, the RHS of Eq. (5), over the number of measurement settings $N$. As we show in Corollary 12 in the Supplementary Information, for small numbers $N$ of measurements it is always beneficial to truncate the Hamiltonian in a controlled way. Indeed, we achieve a reduced total estimation error by imposing that the resulting

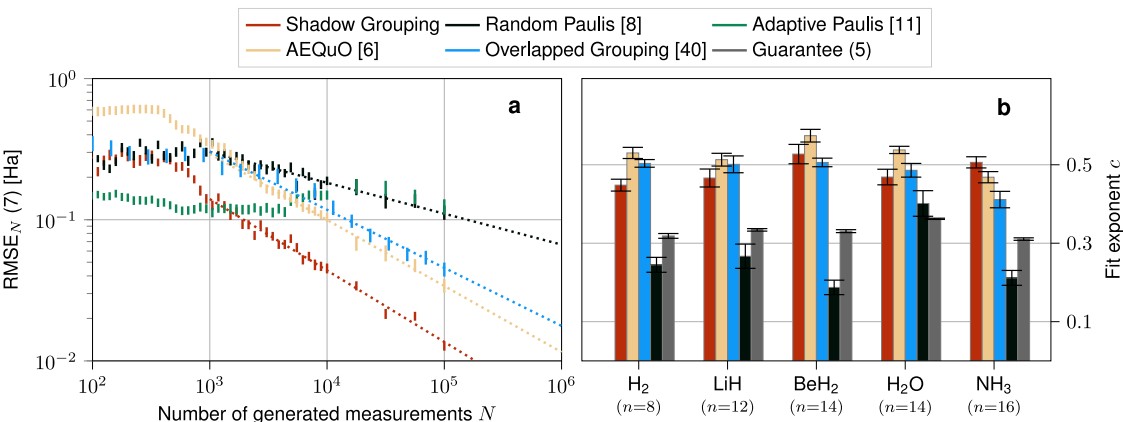

**Fig. 3 | Energy estimation task of the electronic structure problem. a** Empirical accuracies of the various methods for $NH_3$ (JW encoded) as a function of the number of allocated measurements $N$. Deviations of the respective energies are reported as the RMSE (7) in units of Ha. Chemical accuracy is reached below a value of 1.6 mHa. The error bars indicate the empirical standard deviation of each data point. The various methods of the benchmark are discussed in the main text. The data for Derandomization and our tail bound lie above the chart. If possible, each respective data has been fitted according to Eq. (8) (dotted lines). **b** The corresponding exponent, including fit uncertainties for all probed molecules. All molecules have been encoded using the STO-3G minimal basis set except for $H_2$ for which we have chosen the 6-31G encoding. $n$ indicates the number of qubits required to encode the respective molecule.

systematic error is smaller than the expected statistical error. Importantly, this figure of merit can be efficiently calculated given a measurement scheme that generates a list of measurement settings. If a measurement scheme samples the settings, we average the figure of merit over $N_{runs} = 100$ repetitions.

We show the empirical quality measure for the various methods exemplarily for $NH_3$ (using the JW encoding) in Fig. 3. This measure typically decreases following a power law of the form

$$\epsilon(N) = \frac{A}{N^c} , \qquad (8)$$

as indicated by dotted lines underlining the data points. The same qualitative behavior is also observed for the BK and the Parity mapping which we opted to shift to Supplementary Fig. 2 for ease of presentation. The various values for the fit parameter $c$ in Eq. (8) are also provided for all considered molecules and fermion-to-qubit mappings, indicating a robust applicability to a wide range of system sizes $n$. The corresponding plots for the comparison of guarantees have also been shifted to Supplementary Fig. 4. Therein, Overlapped Grouping and ShadowGrouping yield the best results. This is not entirely reflected by looking at the corresponding RMSE: out of the two, only ShadowGrouping yields competitive results for all problem instances. For some of the methods (Derandomization and Adaptive Paulis), increasing the measurement budget $N$ does not even improve the quality of the energy estimation. By further investigating these methods, we find that they typically solely focus on the most dominant terms in the Pauli decomposition which leads to sufficient accuracies related to small measurement budgets. However, they almost completely omit the other terms which are required to improve the measurement accuracy further. For chemical use cases where one frequently requires at least chemical accuracy of $\epsilon_{acc.}^{chem.} \approx$ 1.6 mHa[54], these methods cannot be expected to deliver sufficiently accurate estimations with reasonable measurement effort. This situation is most severe for Derandomization where usually the same measurement setting is generated over and over again. This effect can be mitigated to some extent by appropriately tuning its hyperparameter, but the underlying issue prevails, requiring a case-by-case optimization.

In order to provide a comprehensive comparison of these methods for several molecules, we repeat the fitting procedure of $NH_3$ for the other molecules and fermion-to-qubit mappings in our benchmark. Since we are interested in measuring the energy with an accuracy of at least $\epsilon_{acc.}^{chem.}$, we extrapolate the fits to this accuracy level. The resulting total number of measurement rounds to yield $\epsilon_{acc.}^{chem.}$ is

$$N = \left(\frac{A}{\epsilon}\right)^{1/c}$$

$$\Delta N = \left|\frac{\partial N}{\partial A}\right| \Delta A + \left|\frac{\partial N}{\partial c}\right| \Delta c = \frac{N}{c}\frac{\Delta A}{A} + N \log(N)\frac{\Delta c}{c} , \qquad (9)$$

where the uncertainty $\Delta N$ has been propagated from the uncertainties of the fit parameters. We present the values obtained for the JW-encoding from fitting the respective RMSEs in Fig. 4. Over this range of small molecules, only ShadowGrouping and AEQuO yield a reliable and competitive result. We find that for the smaller problem instances Overlapped Grouping yields competitive results but crucially performs worse at instances of larger system sizes. In particular, for the largest system size in our benchmark, i.e., for $NH_3$, only AEQuO and ShadowGrouping yield reasonable results at all. Here, ShadowGrouping improves upon AEQuO by roughly an order of magnitude. Since the latter includes empirical covariance information to guide the grouping procedure, focussing solely on the coefficients of the observables and their respective commutation relations seems as the striking feature. In contrast to grouping schemes, ShadowGrouping does not impose a fixed clique covering but can implicitly pick from the set of all the suitable cliques by virtue of our tail bound (while never explicitly constructing it). In fact, we have tried to include the empirical variance information into the weighting function weight of ShadowGrouping, but this did not change the generated measurement settings.

Finally, we investigate whether our extrapolated measurement budgets actually ensure a reliable measurement procedure to within $\epsilon_{acc.}^{chem.}$. To this end, we increase the respective extrapolated measurement budgets roughly by a factor of three and run ShadowGrouping to generate suitable measurement settings. We then carry out the measurement to yield an estimate $\hat{E}$ and repeat for $N_{runs} = 100$ independent times. The resulting accuracies for each molecule and fermion-to-qubit mapping are presented in Fig. 4 as colored bars (indicating the standard deviation) around black dots representing the average accuracy. We superimpose these bars with the respective predictions (including

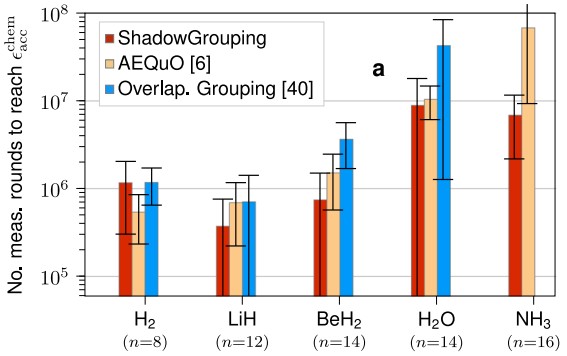
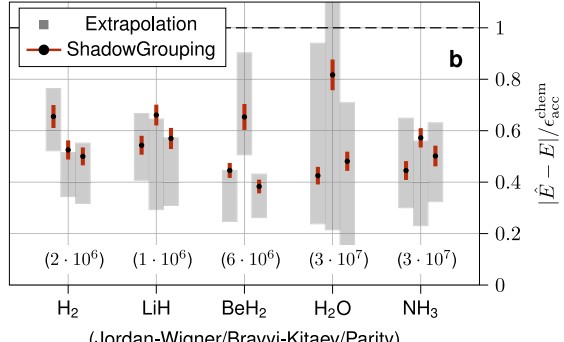

**Fig. 4 | Reliably reaching chemical accuracy $\epsilon_{acc.}^{chem}$ using ShadowGrouping.**
**a** Extrapolated number of measurements $N_{acc.}^{chem}$ to reach chemical accuracy $\epsilon_{acc.}^{chem} = 1.6$ mHa according to Eq. (8) (with uncertainties following Eq. (9)) fitted to the corresponding data from Fig. 3 for the JW-encoding. Only ShadowGrouping and AEQuO reliably yield competitive results (we do not show data if $N_{acc.}^{chem} > 10^8$ or if the fitting procedure did not converge) for all considered instances. **b** For each molecule and the three fermion-to-qubit mappings in our benchmark, we have increased

the total measurement budget $N$ well above the predictions from the extrapolation (respective values for $N$ given in parentheses within the plot) and employed ShadowGrouping to find suitable measurement settings. The colored bars indicate the average accuracy (given in terms of $\epsilon_{acc.}^{chem}$) over 100 independent repetitions. They are superimposed with the corresponding prediction (including its uncertainty (10)) in terms of the gray boxes. In all considered cases, chemical accuracy (dashed line) is reliably reached, and the validity of the extrapolation is verified.

uncertainties

$$\Delta\epsilon = \frac{\Delta A}{N^c} + \epsilon \log N \Delta c \qquad (10)$$

propagated from the fit procedure) obtained from the extrapolation as gray boxes. In all cases, we reliably reach the predicted accuracy. Most importantly, we obtain chemical accuracy in all considered cases. As we show in the Methods subsection 'Energy estimation beyond the ground state', we obtain similar accuracy levels for quantum states beyond the ground state as well, rendering ShadowGrouping an efficient protocol for the energy estimation task of arbitrary quantum states.

## Discussion

Our work achieves two things: first and foremost, we provide rigorous sampling complexity upper bounds for current state-of-the-art readout schemes applicable to near-term feasible quantum experiments, required for the direct energy estimation of quantum many-body Hamiltonians. This marks a shift in paradigm because it enables to formulate unconditional readout guarantees that do not rely on the assumption of having a sufficient proxy of the estimator's variance. Secondly, we propose an efficient and straightforward measurement protocol. It is immediately applicable to VQAs, where the measurement effort is a critical bottleneck. Moreover, our efficient readout strategy is also promising for analog quantum simulators. More generally, it applies to any experiment where a direct measurement of a quantum many-body observable is needed. Technically, our protocols rely on assigning each of the contributions of the Hamiltonian (2) a corresponding weight, which we obtain from probability-theoretic considerations. To this end, we have derived an upper bound to the probability in Theorem 3 that a given empirical estimator fails to yield an $\epsilon$-accurate value for the compound target observable such as the quantum state's energy. This readily provides worst-case ranges for the sample complexity which is useful, for example, in order to appraise the feasibility of employing quantum devices to quantum chemistry problems where accuracy is most crucial[54].

Finally, there are several promising further research directions:

- Investigating the interplay of how the compounds of an observable contribute to the sample complexity, likely based on power-mean inequalities[55], is useful for obtaining tighter bounds for it.
- We do not impose further information on the prepared quantum state $\rho$—this, however, can be used to improve the sample

complexity, e.g., when considering pure states[56] or to incorporate available prior variance estimates[57].

- We have provided guarantees for the energy measurement of a single state. Yet, in VQAs the gradient, i.e., the calculation of the joint energy of multiple states, also has to be calculated with high precision to produce a promising trial state in the first place. Comparing our results to alternative guarantees devised for this problem setting such as refs. 58,59 may be useful to adapt ShadowGrouping to it.
- Our tail bound (4) does not rely on an independent sampling procedure but is compatible with popular grouping schemes. However, in the proof we completely discard any information on the actual empirical variances of the estimates. Refining the upper bound such that it takes into account the empirical variances simultaneously obtained is an exciting question for further investigation as the numerical benchmarks suggest that the grouped mean estimator yields more accurate estimates compared to other estimators. Since ShadowGrouping finds the measurement settings sequentially, the algorithm could easily benefit from such an empirical tail bound as the measurement outcomes can readily be fed back to it. Moreover, this recurs when estimating the covariances of grouped observables to further assess which observables are suited to be measured jointly.
- As ShadowGrouping appears to efficiently provide state-of-the-art groupings based on QWC (see Fig. 3), a numerical benchmark for general commutativity relations (or relations tailored to the hardware constraints[60]) is straightforward and enticing for deeper measurement circuits, appearing to rival QWC even in the presence of increased noise[61]. This decreases the measurement overhead efficiently, especially important for larger system sizes where the number of terms in the decomposition increases rapidly.
- Currently, we are developing a fermionic version of Shadow-Grouping in order to make it more amenable to applications from quantum chemistry.

During the completion of this manuscript, another state-of-the-art scheme has been presented in ref. 62. The numerical benchmark shows that ShadowGrouping is similarly accurate while being computationally more efficient.

## Methods

This section provides further background information on classical shadows that yield the energy estimator (3) as well as details

for replicating the numerical benchmark. We also provide proof sketches for Theorem 3 but refer to the Supplementary Information for detailed proofs. Lastly, we conclude with further details on our algorithm ShadowGrouping such as the motivation for our choice of the weight function. This also includes an examination of our results in light of prior ideas presented in ref. 10 and a comparison to a conceptually easier single-shot estimator.

## Classical shadows

The framework of classical shadows allows us to rewrite the expectation value $o = \text{Tr}[O\rho]$ which we want to estimate in terms of those random variables accessible in the experiment[7,8]. To this end, consider any measurement setting $Q \in \mathcal{P}^n$ that is QWC-compatible with $O$, see Definition 2. Given a state $\rho$, this produces a bit string $\boldsymbol{b} \in \{\pm 1\}^n$ with probability $\mathbb{P}[\boldsymbol{b}|Q, \rho]$. These bit strings contain information about the target observable $O$ as it is compatible with $Q$. Concisely, we have that

$$o = \mathbb{E}_{\boldsymbol{b}} \prod_{i:O_i \neq 1} b_i = \sum_{\boldsymbol{b} \in \{\pm 1\}^n} \mathbb{P}[\boldsymbol{b}|Q, \rho] \prod_{i:O_i \neq 1} b_i. \tag{11}$$

Using Monte-Carlo sampling, this expectation value is estimated by the empirical estimator

$$\hat{o} = \frac{1}{N} \sum_{j=1}^{N} \prod_{i:O_i \neq 1} \hat{b}_i^{(j)} \tag{12}$$

with $\hat{b}^{(j)}$ being the $j$-th bit string outcome of measuring with setting $Q$. This assumes that we have at least $N \geq 1$ compatible measurement settings with the target observable. If not, we can set the estimator equal to 0 and introduce a constant systematic error of at most

$$\epsilon_{\text{syst}}^{(O)} = |h_O|, \tag{13}$$

where $h_O$ may be a corresponding coefficient, e.g., as in Eq. (2). Since Eq. (12) only includes the qubits that fall into the support of $O$, we are not restricted to a single choice of the measurement setting $Q$ as long as $O$ is not fully supported on all qubits. In fact, any measurement setting that is compatible with $O$ is suited for the estimation. Randomized measurements exploit this fact. Strikingly, these random settings come equipped with rigorous sample complexity bounds. For instance, using single-qubit Clifford circuits for the readout we require a measurement budget of

$$N = O\left(\frac{\log(M/\delta)}{\epsilon^2} \max_i 3^{k_i}\right) \tag{14}$$

to ensure a collection of observables $(O^{(i)})_i$ to obey

$$|\hat{o}^{(i)} - o^{(i)}| \leq \epsilon \quad \forall i \in [M] \tag{15}$$

with confidence at least $1 - \delta$, where $k_i$ is the weight of the observable $O^{(i)}$, i.e., its number of non-identity single-qubit Pauli operators[10]. We provide further information on extensions of this method in the Supplementary Information.

## Proof sketch of Theorem 3

In order to derive the energy estimation in confidence bound, we first prove a useful intermediate result which may be of independent interest: a Bernstein inequality for random variables in a Banach space. For this purpose, we extend inequalities from refs. 45,46. In particular, we explicitly extend the vector Bernstein inequality of ref. 45[Theorem

12] to random variables taking values in separable Banach spaces following ref. 46. We call them $B$-valued random variables henceforth. Then, we apply it to random vectors equipped with the 1-norm. A suitable construction of these random vectors finishes the proof of the theorem.

We start by defining $B$-valued random variables following ref. 46 [Chapter 2.1]:

**Definition 5.** Let $B$ be a separable Banach space (such as $\mathbb{R}^n$) and $||\cdot||_B$ its associated norm. The open sets of $B$ generate its Borel $\sigma$-algebra. We call a Borel measurable map $X$ from some probability space $(\Omega, \mathcal{A}, \mathbb{P})$ to $B$ a $B$-valued random variable, i.e., taking values on $B$.

In the Supplementary Information, we show that the norm of the sum of $B$-valued random variables concentrates exponentially around its expectation if we have some information about the variances of the random variables[63]. We summarize this finding by the following $B$-valued Bernstein inequality:

**Theorem 6.** Let $X_1, \ldots, X_N$ be independent $B$-valued random variables in a Banach space $(B, ||\cdot||_B)$ and $S := \sum_{i=1}^{N} X_i$. Furthermore, define the variance quantities $\sigma_i^2 := \mathbb{E}[||X_i||_B^2]$, $V := \sum_{i=1}^{N} \sigma_i^2$, and $V_B := (\sum_{i=1}^{N} \sigma_i)^2$. Then, for all $t \leq V/(\max_{i \in [N]} ||X_i||_B)$,

$$\mathbb{P}\left[||S||_B \geq \sqrt{V_B} + t\right] \leq \exp\left(-\frac{t^2}{4V}\right). \tag{16}$$

As an important corollary, we find that for the Banach space $B = \mathbb{R}^d$ equipped with the $p$-norm ($||\cdot||_B \equiv ||\cdot||_{\ell_p}$) with $p \in [1, 2]$[64] we can tighten the value of $\sqrt{V_B}$ in Eq. (16) to yield the following vector Bernstein inequality:

**Corollary 7.** Let $X_1, \ldots, X_N$ be independent, zero-mean random vectors in $(\mathbb{R}^d, ||\cdot||_{\ell_p})$, $S = \sum_{i=1}^{N} X_i$, and $p \in [1, 2]$. Furthermore, define the variance quantities $\sigma_i^2 := \mathbb{E}[||X_i||_{\ell_p}^2]$ and $V := \sum_{i=1}^{N} \sigma_i^2$. Then, for all $t \leq V/(\max_{i \in [N]} ||X_i||_{\ell_p})$,

$$\mathbb{P}\left[||S||_{\ell_p} \geq \sqrt{V} + t\right] \leq \exp\left(-\frac{t^2}{4V}\right). \tag{17}$$

This corollary includes the edge case of $p = 2$ proven in ref. 45 [Theorem 12]. Our main finding, Theorem 3, follows by a suitable choice of random vectors. While the proof is included in the Supplementary Information, we want to comment on its implications.

**Remark 8.** The resulting tail bound (4) keeps a balance between the magnitude of the coefficients $h_i$ of each observable and how often they have been measured, respectively. In the following section, we show how to turn this insight into the weight function weight.

**Remark 9.** Due to the inherent commutation relations in Eq. (2), the dependence of the tail bound (4) on the $N_i$ necessarily becomes slightly convoluted. In fact, we can identify $||\boldsymbol{h}'||_{\ell_1}^{-2}$ to be proportional to a weighted power mean of power $r = -1/2$ where the mean runs over $(N_i)_i$ with weights $(|h_i|)_i$. Similarly, $||\boldsymbol{h}''||_{\ell_1}$ is inversely proportional to the mean with $r = -1$. Some of these means' properties and relations to other means can be found, e.g., in ref. 55.

To make this more precise, the weighted power mean of $\boldsymbol{x} \in \mathbb{R}^d$ with weights $\boldsymbol{w} \in \mathbb{R}_{\geq 0}^d$ and power $r$ is defined as

$$M_r(\boldsymbol{x}|\boldsymbol{w}) := \left(\frac{\sum_{i=1}^{d} w_i x_i^r}{\sum_{i=1}^{d} w_i}\right)^{1/r}. \tag{18}$$

For non-negative $\boldsymbol{x}, \boldsymbol{w}, M_r$ is monotonously increasing with $r$.

Now, we set $w_i = |h_i|$ and $x_i = N_i$. Thus, we have that

$$M_{-1/2}(\boldsymbol{x}|\boldsymbol{w}) = \frac{\|\boldsymbol{h}\|_{\ell_1}^2}{\|\boldsymbol{h}'\|_{\ell_1}^2} \geq \frac{\|\boldsymbol{h}\|_{\ell_2}^2}{\|\boldsymbol{h}'\|_{\ell_1}^2} \qquad (19)$$

$$M_{1/2}(\boldsymbol{x}|\boldsymbol{w}) = \left(\frac{\sum_i |h_i|\sqrt{N_i}}{\|\boldsymbol{h}\|_{\ell_1}}\right)^2 \qquad (20)$$
$$\leq \frac{\|\boldsymbol{h}\|_{\ell_2}^2}{\|\boldsymbol{h}\|_{\ell_1}^2}\sum_i N_i \,,$$

due to Cauchy's inequality. Then, by $M_r$ monotonously increasing with $r$,

$$\|\boldsymbol{h}'\|_{\ell_1} \geq \frac{\|\boldsymbol{h}\|_{\ell_1}}{\sqrt{\sum_i N_i}} \qquad (21)$$

follows as a lower bound. In the limiting case of non-commuting observables, this bound reduces to $\|\boldsymbol{h}\|_{\ell_1}/\sqrt{N}$ where $N$ is the total shot number. The converse, i.e., extracting upper bounds is not as straightforward to do.

## Finding an equivalent weight function for the energy estimation inconfidence bound

In this section, we find an equivalent weight function weight for our tail bound (4) that can be used to assess each observable's individual contribution to the total bound. As of now, the current bound depends on all contributions jointly. However, the 1-norm in Eq. (4) makes a subdivision into the individual contributions possible. To this end, we inspect the bound further. We start with a further upper bound[65][Theorem 2.4] to Eq. (4) to get rid of the mixed terms in the square and conclude that

$$\mathbb{P}\left[|\hat{E} - E| \geq \epsilon\right] \leq \exp\left(-\frac{1}{4}\left[\frac{\epsilon}{2\|\boldsymbol{h}'\|_{\ell_1}} - 1\right]^2\right) \qquad (22)$$
$$\leq \exp\left(-\frac{\epsilon^2}{32\|\boldsymbol{h}'\|_{\ell_1}^2} + \frac{1}{4}\right) < 1.3 \exp\left(-\frac{\epsilon^2}{32\|\boldsymbol{h}'\|_{\ell_1}^2}\right)$$

is to be minimized the most when trying to minimize the failure probability. Since the argument is monotonously increasing with $\|\boldsymbol{h}'\|_{\ell_1}$, this is equivalent to minimizing $\|\boldsymbol{h}'\|_{\ell_1} = \sum_i |h_i|/\sqrt{N_i}$. In order to decrease the sum the most we need to find a measurement setting such that the summands change the most. If we define

$$w_i := |h_i|\left(\frac{1}{\sqrt{N_i}} - \frac{1}{\sqrt{N_i+1}}\right) = |h_i|\frac{\sqrt{N_i+1} - \sqrt{N_i}}{\sqrt{N_i(N_i+1)}} > 0 \,, \qquad (23)$$

since $N_i \geq 1$ as per Theorem 3, the optimization boils down to maximizing $\sum_i w_i > 0$. Therefore, we come back to a form of the objective function where the arguments of the sum serve as the individual weights for each of the observables in the Hamiltonian. As a consequence, we can readily provide the weights to ShadowGrouping as sketched in Fig. 2. There is one caveat left: Theorem 3 does not hold if any of the Pauli observables has no compatible measurements, i.e., in case of $N_i = 0$ for some $i$. In this case, Eq. (23) is ill-defined. Using the fact that $w_i \leq |h_i| \forall i$, we can numerically rectify the issue by setting

$$w_i = \alpha|h_i| \quad \text{if} \quad N_i = 0 \qquad (24)$$

with some hyperparameter $\alpha \geq 1$ that balances the immediate relevance of terms that have no compatible measurement setting yet with those that do but are of larger magnitude $|h_i|$ in the Hamiltonian decomposition. Repeated rounds eventually lead to all Pauli observables having at least one compatible measurement setting such that we can evaluate Eq. (4). In our numerics, we choose $\alpha$ such that

observables with no compatible measurement setting yet are always preferred over the ones that do. Setting

$$h_{\min} := \min_i|h_i| > 0$$
$$h_{\max} := \max_i|h_i| > 0 \,, \qquad (25)$$

we find $\alpha$ to be at least

$$\alpha > \frac{h_{\max}}{h_{\min}} \geq 1 \,. \qquad (26)$$

We use $\alpha = h_{\max}/h_{\min} + h_{\min}$ throughout all numerical experiments. This way, we scan through all observables at least once to rank them according to Eq. (23) and find suitable settings afterwards. This overhead introduced can be mitigated by applying the truncation criterion, Corollary 12 in the Supplementary Information, and running Shadow-Grouping on the smaller observable set again. This combination is computationally efficient (it roughly doubles the computational overhead) and ensures that only the observables of statistical relevance to the tail bound are considered in the first place.

## Comparison with derandomization

While our tail bound, Theorem 3, is the first derived upper bound for the energy estimation of quantum many-body Hamiltonians with currently feasible readout schemes, there is pioneering work in this direction in ref. 10. Here, a tail bound is found by means of Hoeffding's inequality that at least one unweighted, single Pauli observable in a given collection deviates substantially from its mean. This situation is somewhat related to the task of estimating the energy by summation of many Pauli observables but discards the different weights in the Pauli decomposition as well as their respective interplay as each of the observables are treated independently of the other. The authors of ref. 10 reintroduce the weights in an ad-hoc manner. Hence, we refer to it as the Derandomization bound which originally reads as

$$\text{DERAND}_i^{(\text{orig})} := 2\exp\left(-\frac{\epsilon^2}{2}N_i\frac{\max_j|h_j|}{|h_i|}\right). \qquad (27)$$

Here, $\epsilon$ is again the accuracy, $N_i$ counts the number of previous compatible measurement settings and the $h_i$ come from Eq. (2). Taking into account the weights $h_i$ in an ad-hoc fashion, however, renders this expression unsuitable for an actual upper bound. We rectify this by shifting the parameter $O \mapsto hO =: \tilde{O}$ by some value $h \neq 0$. Hoeffding's inequality implies that

$$\mathbb{P}[|\hat{\tilde{o}} - \tilde{o}| \geq \epsilon] \leq 2\exp\left(-\frac{\epsilon^2}{2h^2}N\right). \qquad (28)$$

This ensures that all weighted observables in Eq. (2) are treated equally w.r.t. the value of $\epsilon$. Thus, the actual Derandomization bound reads as

$$\text{DERAND}_i := 2\exp\left(-\frac{\epsilon^2}{2h_i^2}N_i\right). \qquad (29)$$

Taking a union bound over all observables $O^{(i)}$, we again obtain an upper bound for $\mathbb{P}[|E - \hat{E}| \geq \epsilon]$. We see that this derivation leads to a more conservative $\epsilon$-closeness (captured in terms of the $\infty$-distance) compared to the 1-distance of Theorem 3. Since we treat each observable independently of all the others, the total accuracy of the energy estimation can grow as

$$\epsilon_{\text{eff}} := \left|\sum_{i=1}^M \left(h_i\hat{o}^{(i)} - h_i o^{(i)}\right)\right| \leq \sum_{i=1}^M \underbrace{\left|h_i\hat{o}^{(i)} - h_i o^{(i)}\right|}_{\leq \epsilon} = M\epsilon \qquad (30)$$

**Table 1 | Comparison of our tail bound (4) with the Derandomization bound (29) adapted from ref. 10**

|  | Theorem 3 | Derandomization |
|---|---|---|
| norm | $\ell_1$ | $\ell_\infty$ |
| $\epsilon_{\text{eff}}$ | $\epsilon$ | $M\epsilon$ |
| equation | $\exp\left(-\frac{1}{4}\left[\frac{\epsilon}{2\|\boldsymbol{h}'\|_{\ell_1}}-1\right]^2\right)$ | $2\sum_{i=1}^{M}\exp\left(-\frac{\epsilon^2}{2h_i^2}N_i\right)$ |
| weight | Equation (23) | $c^{N_i}-c^{N_i+1}$ $c=\exp(-\epsilon^2/(2h_i^2))$ |

Norm refers to how the error is captured w.r.t. the single Pauli terms in Eq. (2) whereas $\epsilon_{\text{eff}}$ refers to the error in terms of the energy estimation. For Theorem 3, this is identical to the guarantee parameter $\epsilon$ while the Derandomization guarantee scales with the number of qubits $n$. The difference arises from the fact that we effectively exchange the sum and the exponential function in the corresponding bounds. The latter are used to derive a weight function weight for ShadowGrouping, see the previous section.

via the generalized triangle inequality. Since in typical scenarios $M = \text{poly}(N)$, this implies that the guarantee parameter $\epsilon$ scales with the number of qubits in order to guarantee $|\hat{E}-E| \leq \epsilon_{\text{eff}}$, requiring even more measurement settings to compensate for this effect. We summarize and compare both tail bounds for $|\hat{E}-E| \geq \epsilon$ in Table 1.

We also compare ShadowGrouping to the Derandomization measurement scheme. First, ShadowGrouping does not require a qubit ordering as it directly works with the inherent commutation relations in Eq. (2). The Derandomization algorithm, on the other hand, finds the measurement setting qubit by qubit and thus imposes an ordering of the observables. As a consequence, the computational complexity of our scheme scales with $\mathrm{O}(nM\log(M))$ for assigning a single measurement setting as we have to order the $M$ weights first in descending order, then go through every target observable comprised of $n$ qubits. By contrast, the Derandomization procedure scales as $\mathrm{O}(nM)$ as it has to modify all $M$ terms in its corresponding bound after appending a single-qubit Pauli observable to the next measurement setting. We see that our approach only worsens the scaling by a logarithmic factor but enables the algorithm to find the next measurement setting regardless of the qubit order (the Derandomization procedure always uses the same ordering). This might help to decrease the inconfidence bound quicker. Moreover, the Derandomization scheme requires a continuation of the tail bound to the case of having partially assigned the next measurement setting. This is possible for the Derandomization bound[10] but unclear in case of our tail bound. ShadowGrouping, on the other hand, can be applied to either bound.

**Comparison with single-shot estimator**

We employ the Weighted Random Sampling method of ref. 66 to assess the scaling of our measurement guarantee (5). This estimator simply picks a single observable in the Hamiltonian decomposition with probability $p_i = |h_i|/\|\boldsymbol{h}\|_{\ell_1}$ and obtains a single-shot estimate. Hence, we refer to it as the single-shot estimator. This way, the state's energy can be estimated from a single measurement round. Importantly, this sampling strategy can be straightforwardly equipped with a guarantee. Assuming, we have picked the $k$-th observable to be measured, we have

$$\hat{E} = s_k \|\boldsymbol{h}\|_{\ell_1}$$
$$s_k := \text{sign}(h_k)\hat{o}^{(k)} \in \{\pm 1\}. \tag{31}$$

This estimator is unbiased:

$$\mathbb{E}[\hat{E}] = \sum_{i=1}^{M} p_i \|\boldsymbol{h}\|_{\ell_1} \mathbb{E}[s_i] = \sum_{i=1}^{M} |h_i| \text{sign}(h_i)\mathbb{E}[\hat{o}^{(i)}] = \sum_{i=1}^{M} h_i o^{(i)} = E.$$

**Table 2 | Accuracy in terms of the RMSE-metric (7) at a measurement budget of $N = 1000$**

| Molecule $E$ [mHa] | Enc. | Random Paulis[8] | Single shot Equation (31) |
|---|---|---|---|
| H$_2$ $-1.86\times10^3$ | JW | $123\pm15$ | $360\pm40$ |
|  | BK | $114\pm13$ | $300\pm40$ |
|  | Parity | $134\pm16$ | $370\pm50$ |
| LiH $-8.91\times10^3$ | JW | $84\pm10$ | $380\pm40$ |
|  | BK | $92\pm10$ | $340\pm40$ |
|  | Parity | $97\pm12$ | $350\pm40$ |
| BeH$_2$ $-19.05\times10^3$ | JW | $170\pm18$ | $640\pm70$ |
|  | BK | $158\pm22$ | $610\pm70$ |
|  | Parity | $130\pm16$ | $620\pm80$ |
| H$_2$O $-83.60\times10^3$ | JW | $320\pm40$ | $1980\pm220$ |
|  | BK | $430\pm50$ | $2030\pm280$ |
|  | Parity | $670\pm70$ | $1980\pm240$ |
| NH$_3$ $-66.88\times10^3$ | JW | $430\pm50$ | $2000\pm230$ |
|  | BK | $340\pm40$ | $2170\pm250$ |
|  | Parity | $470\pm50$ | $2060\pm210$ |
| Values above in mHa |  |  |  |

The mean over $N_{\text{runs}} = 100$ independent repetitions including its standard deviation are reported. For reference, the ground-state energies $E$ for each molecule are also provided. The single-shot estimator defined in Eq. (31) does not produce competitive estimates even when benchmarked against random Paulis employed in Fig. 3.

Clearly, $|\hat{E}| \leq \|\boldsymbol{h}\|_{\ell_1}$. Invoking Hoeffding's inequality, for $N$ many independent samples, we have that

$$\mathbb{P}[|\hat{E}-E| \geq \epsilon] \leq 2\exp\left(-\frac{N\epsilon^2}{2\|\boldsymbol{h}\|_{\ell_1}^2}\right) \tag{32}$$

given some $\epsilon > 0$. We arrive at a sample complexity (with $\delta \in (0,1/2)$) of

$$N \geq \frac{2\|\boldsymbol{h}\|_{\ell_1}^2}{\epsilon^2}\log\frac{2}{\delta} \tag{33}$$

with probability $1-\delta$ in order for $|\hat{E}-E| \leq \epsilon$. Solving for $\epsilon$, we compare this guarantee to Eq. (5). With $\log(2/x) \leq 2\log(1/x)$ for $x \leq 1/2$, we have

$$\epsilon_{\text{single}} = \sqrt{2}\sqrt{\log\frac{2}{\delta}}\frac{\|\boldsymbol{h}\|_{\ell_1}}{\sqrt{N}} \leq 2\sqrt{2\log\frac{1}{\delta}}\sum_{i=1}^{M}\frac{|h_i|}{\sqrt{N}} \leq 2\sqrt{2\log\frac{1}{\delta}}\sum_{i=1}^{M}\frac{|h_i|}{\sqrt{N_i}}$$

$$\leq \alpha_\delta\|\boldsymbol{h}'\|_{\ell_1} \equiv \epsilon_{\text{multi}}$$

$$\Rightarrow \tilde{\mathrm{O}}(\epsilon_{\text{single}}) = \tilde{\mathrm{O}}(\epsilon_{\text{multi}}),$$

with $\alpha_\delta$ from Supplementary Eq. (31), see also the proof of Corollary 12 in the Supplementary Information. We find that the two guarantees agree up to logarithmic factors in $N$. Moreover, in case all observables commute with each other, both tail bounds are equivalent up to a constant factor. However, in the numerical benchmark within the corresponding Results' subsection, we see that the estimator (31) does not fare better than the random Pauli settings, see Table 2. We attribute this discrepancy to the fact that the grouped mean estimator bears a lower variance in practice than the single-shot estimator introduced here. It implies that recycling the measurement outcomes is the more favorable approach and hints towards a possible refinement of our tail bound.

**Energy estimation beyond the ground state**

We investigate ShadowGrouping's estimation capabilities beyond the ground state. We do so in two separate ways. In the first instance, we gradually increase the mixedness of the quantum state to be measured

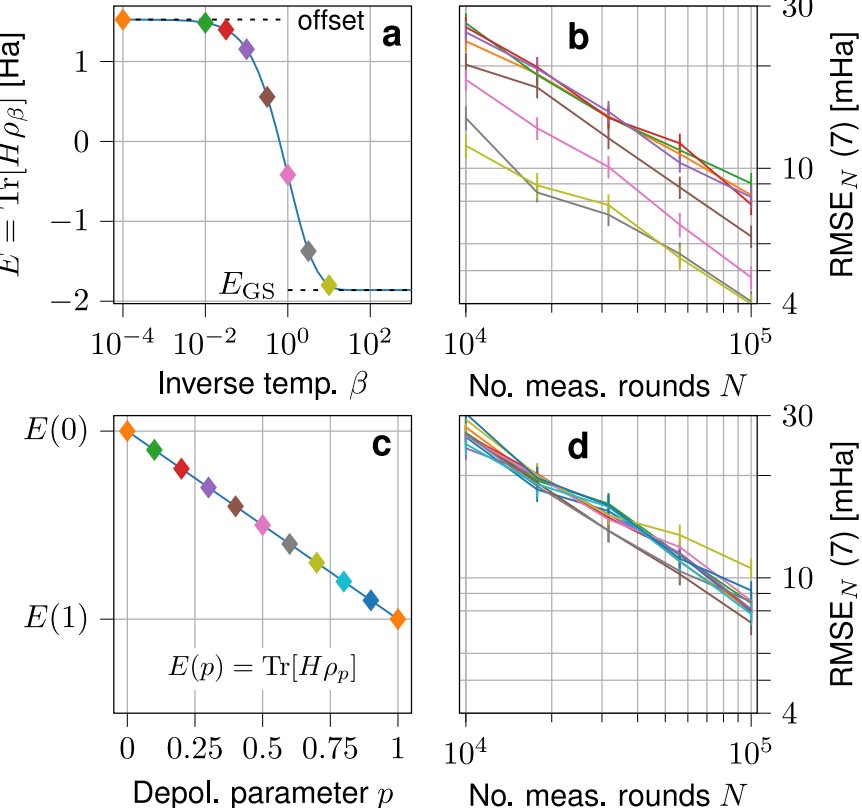

**Fig. 5 | Empirical accuracy of ShadowGrouping for the energy estimation of states beyond the ground state.** Energy of the thermal state (34) for various values of the inverse temperature $\beta$ (**a**). Indicated are eight states whose energy has been estimated by ShadowGrouping. The respective average RMSE as a function of the total number of measurements is shown for the JW-mapping (**b**). Error bars indicate the standard deviation over 100 repetitions. **c, d** energy and estimation accuracy profiles of a depolarized Haar-random state $\rho_p$ (35) with depolarization parameter

$p$. The eleven probed states range from a pure state to the maximally mixed state and from states with structure (the ground state) to states with little structure (Haar-random states). In all cases, the observed accuracy levels are of the same order of magnitude. Moreover, even the arguably worst-case state, i.e., the maximally mixed state of largest variance, only increases the reached accuracy level by a constant factor of less than three.

by considering the thermal state

$$\rho_\beta = \frac{e^{-\beta H}}{Z}, \tag{34}$$

with inverse temperature $\beta$ and partition function $Z$ and the Hamiltonian from Eq. (2) for the $H_2$-molecule in 6-31G encoding (8 qubits). The ground state is contained in this state class for the limit of $\beta \to \infty$. On the other hand, the maximally mixed state $\mathbb{1}/2^n$ is attained for $\beta \to 0$, allowing for a smooth interpolation between the structured pure state and the unstructured mixed one. We show the energy $E(\beta) = \text{Tr}[H\rho_\beta]$ in Fig. 5. We pick eight different values from the most relevant range for $\beta$ and prepare the measurement settings by ShadowGrouping and with a measurement budget of $N = 10^5$. Since ShadowGrouping works deterministically, we only need to do this routine once and reuse the resulting settings for any subsequent measurement procedure. We track the RMSE (7) over 100 independent measurement repetitions. Because the mixedness of the quantum state increases its energy variance, we find that the observed estimation error increases slightly with smaller $\beta$. Nevertheless, the overall error follows the same scaling with $N$ and is only a constant factor of less than three worse. The plots for the BK and Parity mapping qualitatively behave the same as shown in Supplementary Fig. 3 in the Supplementary Information.

In the second instance, we directly consider quantum states with no inherent structure by drawing them Haar-randomly, i.e., we draw state vectors uniformly from the complex sphere. To this end, we generate such a Haar-random state $|\psi\rangle$ with energy $E(p=0) = \langle\psi|H|\psi\rangle$.

When averaged over all Haar-random states, this value will vanish. However, we consider estimating the correct energy of a single state for now. Again, we smoothly interpolate between the pure state and the maximally mixed state by means of global depolarization noise with parameter $p$. The resulting state thus becomes

$$\rho_p = (1-p)|\psi\rangle\langle\psi| + \frac{p}{2^n}\mathbb{1}. \tag{35}$$

We carry out the same analysis as for the thermal state (we use a different Haar-random state for each of the 100 repetitions) and present the results in Fig. 5. Since Haar-random states do not possess any structure, we find a similar quantitative accuracy level for all probed values of $p$. We therefore conclude that ShadowGrouping yields a comparable performance (up to a state-dependent variance factor) for arbitrary quantum states, in line with our guarantees of Theorem 3 that hold uniformly for all quantum states as well.

## Data availability

The Hamiltonian decompositions used for the benchmarks in Figs. 3, 4, Table 2 and Supplementary Figs. 2–4 have been sourced from an online repository[67]. Resulting data (such as Eq. (7) as a function of $N$ or the measurement settings produced by ShadowGrouping) generated for the benchmarks is stored in ref. 52. These benchmark data generated in this study have been deposited in a git-repository free of any accession code [https://gitlab.com/GreschAI/shadowgrouping/-/blob/master/data.zip?ref_type=heads].

## Code availability

All computer code required to reproduce Figs. 3–5 and Table 2 as well as Supplementary Figs. 1–4 either from intermediate data or from scratch have been deposited in ref. 52.

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

## Acknowledgements
We would like to thank Zihao Li and Bruno Murta for feedback on the presentation of our proofs. This work has been funded by the Deutsche Forschungsgemeinschaft (DFG, German Research Foundation) via the Emmy Noether program (Grant No. 441423094) (A.G. and M.K.); the German Federal Ministry of Education and Research (BMBF) within the funding program Quantum technologies—From basic research to market in the joint project MANIQU (Grant No. 13N15578) (A.G. and M.K.); and by the Fujitsu Germany GmbH and Dataport as part of the endowed professorship Quantum Inspired and Quantum Optimization (A.G. and M.K.).

## Author contributions
A.G. carried out all calculations and numerical studies, M.K. supervised the process and conceived the idea of applying the vector Bernstein inequality. Both authors wrote the manuscript together.

## Funding

## Competing interests
The authors declare no competing interests.
