## [Transparent Peer Review file · Nature Communications]

Guaranteed efficient energy estimation of quantum many-body Hamiltonians using ShadowGrouping

Corresponding Author: Professor Martin Kliesch

Version 0:

Reviewer comments:

Reviewer #1

(Remarks to the Author)
Please find it in the attachment

Reviewer #2

(Remarks to the Author)

The paper "Guaranteed efficient energy estimation of quantum many-body Hamiltonians using ShadowGrouping" by Gresch and Kliesch describes a new technique for using limited numbers of shots to obtain the highest provable accuracy within methods such as the variational quantum eigensolver. By using shadow estimation methods, the authors are able to provide both a provable bound and a numerically practical method. The algorithm is well described and the general problem - minimizing measurements - is an important one to solve. However, the provable bound shows an unpractical number of required measurements and the numerical results do not do enough to convince the reader the ShadowGrouping is optimal. For the broad readership of Nature Communications, I believe that there needs to be significantly more clearly defined demonstration that ShadowGrouping is optimal. As it stands now, the results are of interest to a smaller community. As such, I cannot accept the paper unless major revisions can more conclusively demonstrate ShadowGrouping's benefits. I have specifically pointed out, in the Major Comments section below, the need for at least more numerics. I also list some Minor Comments and Typos that should be addressed.

* Major Comments

- As pointed out by the authors, the worst-case bounds are not particularly practical. During a typical VQE run, many states, of varying variances, will be seen. It's certainly true that the worst case bounds are unlikely to be seen, but understanding a more practical example is necessary to demonstrate ShadowGrouping's impact. The authors provide some evidence that ShadowGrouping does perform numerically well in the Table I: Empirical Benchmarks. However, none of the results presented in Table I, ShadowGrouping or otherwise, approach chemical accuracy. They are in fact one or two orders of magnitude away for the fixed shot budget of $N=1000$. Given the goal of 1.6mHa accuracy, and the results shown in Figure 5 for worst-case measurements required for chemical accuracy, I think that the authors should perform additional numerics to find the 'real-world' number of shots required for chemical accuracy. A numerical study showing the energy error as a function of number of shots, going from the 1000 to whatever is required for chemical accuracy (10^6 ? 10^{10} ?) would greatly strengthen the paper. This would answer a lot of lingering questions about ShadowGrouping. Is it consistently better than other methods at different numbers of shots? Is the real-world number of shots significantly lower than the worst case bounds? How much better is ShadowGrouping at convergence? I think the data over many numbers of shots is also useful, as there are proposals for using an adaptive number of shots during optimization runs (see <https://quantum-journal.org/papers/q-2020-05-11-263/> or <https://quantum-journal.org/papers/q-2023-03-16-949/> for examples).

* Minor Comments

- Page 7, table I caption. The caption references "numbers in parentheses". There seem to be no parentheses in the table (except for a reference to equation (9)). Perhaps this text is erroneous?

- The authors should consider adding the number of qubits used for each molecule, explicitly - this is implicit in the basis definition, but that conversion is not 'common knowledge'.

* Typos

- Page 1, Line 58: "one aim to only..." maybe should be "one aims to only"?
 - Page 3, line 25: "many measurements round in which" maybe should be "many measurement rounds in which"?
 - Page 3, line 75, Definition 1: "Let H a Hamiltonian as in Eq. (2)" maybe should be "Let H be a Hamiltonian"? Or maybe some other verb / symbol is missing?
 - Page 6, line 66: "without any additionally two-qubit..." should probably be "without any additional two-qubit"
 - Page 7, line 7: "against state-of-the-art method from..." should probably be "against state-of-the-art methods from..."
 - Page 8, line 32: "the read-out of read-out of quantum.." should probably be "the read-out of quantum experiments..."
 - Page 9, line 72: "classical shadows allows use to rewrite..." should probably be "classical shadows allows us to rewrite..."
 - Page 10, line 14: Perhaps there is a period missing? "... Banach spaces following Ref. [45] We call" should maybe be "... Banach spaces following Ref. [45]. We call"
- XS

Version 1:

Reviewer comments:

Reviewer #1

(Remarks to the Author)

Dear Editor Marco Bentivegna,

thank you for having considered me as a reviewer for this second round. After reading the Authors' response, my opinion is unchanged. I will not address point-by-point the Authors' response, as my review would look very similar to the first one. The disagreement mostly stem from fundamental assumptions/axioms, such as whether one should employ the ACTUAL error, which is unaccessible, or an ESTIMATED error, which can be accessed. Based on my first round of review and the Authors' answers, I believe that you, Editor, can take a well informed decision on whether the assumptions on which the Authors' algorithm is based are appropriate (or not) for a measurement protocol that will be employed in real, large scale experiments.

Also, I do believe that the comparison with other approaches is not very precise. I am not speaking only of AEQuO, whose performance is still surprisingly poor. As can be seen in Fig. 3 of the revised manuscript, the Derand and Overlapped Grouping (OG) methods reach horizontal plateaus. In the case of the Derand, I believe that this is due a rounding error in the code, that occurs for large numbers of measurements. In this case, the Derand code starts allocating measurements to the XXX...XXX clique only, which would explain the horizontal plateau. In the case of the OG protocol, I believe that the authors do not appropriately set a very hidden parameter inside the OG code, that determines how many and which Paulis are to be disregarded in the estimation process. This explains the plateau since, when enough many measurements are allocated, these Pauli starts contributing to a shift away the exact average.

These two (plausible) imprecisions are certainly not the Authors' fault, yet still make their benchmark not very accurate.

Thanks, and my best regards,
Luca Dellantonio

(Remarks on code availability)

Reviewer #2

(Remarks to the Author)

The authors have performed many additional numerical experiments to address my comments from the previous review. Namely, they performed additional numerical tests up until 10^5 shots and then extrapolated to get the number of shots required for chemical accuracy. This certainly increases confidence in this work and demonstrates that the ShadowGrouping method should generally out perform the other methods. However, I wonder why the authors decided to extrapolate, rather than directly perform the experiments, up until 10^7 or 10^8 shots. It is certainly nice to have the exponent c , but it would be nice to see ShadowGrouping actually achieve chemical accuracy for the largest system. While I understand that this will be at least 100x more costly than the experiments already performed, this gives the appearance of a bottleneck somewhere. Systems that are much larger than these are the ones that we really care about, where we would be running directly on a quantum computer with, say, 100+ qubits. A really, really rough extrapolation from H₂ (8 qubits, around 2×10^5 expected shots) to NH₃ (16 qubits, around 5×10^6 shots) would say the number of shots goes up by more than an order of magnitude when the number of qubits doubles. Will ShadowGrouping be able to perform at 10^9 shots? I think, with minor revisions, this manuscript could be accepted into Nature Communications. Whether this is through an additional simulation explicitly showing ShadowGrouping achieving chemical accuracy, or a discussion about scaling towards larger number of shots, I

leave up to the authors.

(Remarks on code availability)

Reviewer #3

(Remarks to the Author)

See attachments.

(Remarks on code availability)

Version 2:

Reviewer comments:

Reviewer #2

(Remarks to the Author)

- The authors have performed additional numerical experiments, pushing the number of shots substantially higher to provide the numerical proof of chemical accuracy and provided additional detail about scalability. This closes the final issue that I had with the paper, and, thus, I would accept the paper. Looking at the comments from Reviewer #1, who has a much deeper knowledge of these methods, they still seems to have some reservations. Their comments have, through revisions, greatly increased the quality of the paper, by pointing out fine-technical details about hyperparameters that made the benchmark comparison better - and hopefully more accurate. As Reviewer #1 pointed out, tuning hidden hyperparameters is not the authors' fault, but it seems that the comparisons are now much better than before. It is hard to say whether the best hyperparameters have been chosen, but I believe the authors have provided considerable and respectable effort into providing good faith comparisons sufficient for publication.

As a minor presentation issue, I'm not sure why figure 4a has two color boxes in the legend.

(Remarks on code availability)

Reviewer #3

(Remarks to the Author)

The authors have addressed all of my questions, and I believe the manuscript is now ready for publication in NC. I recommend that the authors conduct a thorough review of the manuscript before submitting the final version. For instance, there is a typo on page 5, second column: "next element from this ordered list For the QWC-case."

(Remarks on code availability)

Reply to Referee 1

We thank Luca Dellantonio for the thoughtful and critical review.

Before going into the discussion of the specific issues we would like to make an overall comment. Our work represents a shift in paradigm for the energy estimation procedure: previous work has aimed at constructing unbiased energy estimators that bear the lowest possible (expected) variance. Here, one very popular and successful approach is the partitioning of the Pauli decomposition into commuting families, called grouping schemes. Within these approaches, the justification stems from Chebyshev's inequality to upper-bound $\mathbb{P}[|\hat{E} - E| \geq \epsilon]$ in terms of $\text{Var}[\hat{E}]$. However, the exact value of the variance is not known and the analysis has to resort to an empirical proxy of the variance. This introduces an additional error, which is not fully rigorously controllable, for a limited number of samples.

With our work, we successfully achieved the goal to extend tail bounds often used in quantum information theory to the energy estimation setting. In this way, we obtain accuracy guarantees that hold unconditionally and which can also be applied to other known methods. For instance, it yields a new guarantee for AEQuO under the usage of empirical variance estimates.

In fact, this step is not trivial and we had to extend tail bounds for random vectors for the here relevant norms to prove our guarantees. However, it does not only provide a guarantee but also is the basis of our practical measurement scheme, which turns out to outperform all state-of-the-art methods we know, including AEQuO.

In the revised manuscript, we explain this paradigm shift (justification via direct convergence of the energy estimator instead of the detour via the convergence of the empirical variance). Moreover, we have taken the criticism of the referee seriously and have fully revised our numerical demonstration. The new numerical results demonstrate more clearly that ShadowGrouping provides a better accuracy than other methods.

We have included versions of our manuscript and Supplementary Information in our resubmission where all changes are highlighted.

Now, we go through the major specific points raised by the referee.

(1) The performance of measurement protocols cannot be tested with the estimator in Eq. (9), as the exact value of the quantity being measured is generally unknown. For a measurement protocol to be meaningful, aside from the estimated average value it must provide a meaningful estimate for the error (not a bound). In our work, we provide the theory to estimate the proper error (see Eqs. (2) in Ref. [3]), that can be adapted to any measurement protocol – including all ShadowGrouping variants. In Table I and for all numerical results, the authors should employ an estimator that is agnostic of the exact value of the observable to be measured. As a side remark, our error estimator in Eqs. (2) from Ref. [3] can be employed to give the statistically exact estimation error without the requirement of repeated iterations that are numerically expensive. This can be very helpful when presenting the numerical results, since statistical fluctuations from repeated measurements can be large. Also, it can come at handy for my criticism number 5 below.

We thank the referee for bringing up this issue. Indeed, for system sizes that are too large for classical simulations, the estimation error needs to be inferred from the measured data. The referee and his co-authors provide a method to obtain an uncertainty $\Delta\hat{E}$ of an energy estimator. As pointed out by the referee below, this method can also be

applied to our estimation protocol.

However, this uncertainty does not quantify the actual error and also does not provide a rigorous bound to it for limited numbers of samples. Tail bounds provide a robust supplement (and not a substitute) to such uncertainty estimates: they catch potential outliers to which estimators, especially in the form of empirical means, are prone. There are two subtle but important differences between the tail bounds and uncertainty estimates. First, for finitely many measurement rounds, the two quantities yield two related, yet different confidence intervals: Uncertainty estimates provide intervals around the current point estimate \hat{E} (which most often does not equal the target energy E) whereas tail bounds provide them around the actual energy E . This is why it is not mathematically rigorous to solely rely on a small variance to infer a small estimation error. Secondly, relying on the variance is neither optimal: since the variance estimator is of higher moment than the mean estimator (which is the primary quantity of interest), it comes with its own convergence rate. This holds in particular if one relies on approximating the covariance with another empirical estimator. In general, this introduces a non-negligible estimation error for finite measurement rounds, cf. Ref. [2, Proposition 4], which is not accounted for in the uncertainty estimate $\Delta\hat{E}$. Tail bounds do not require any assumptions on the (empirical) variance as they aim to suitably bound the quantity of interest, i.e. to upper-bound $|\hat{E} - E|$. Since no convergence assumptions on the (co)variance of the energy estimator are required, the subsequent guarantee thus holds *unconditionally* for finite measurement rounds.

Regarding our numerical experiments, we agree that employing the root mean squared error (RMSE) as it includes the unknown quantity E is not scalable to larger problem instances. Nevertheless, we employed it because of two reasons: first of all, this quantity includes the target of our tail bound, namely $|\hat{E} - E|$. Since ShadowGrouping has been devised to minimize a suitable upper-bound to this quantity, this empirical benchmark, although not scalable, serves as a consistency check whether this optimization is viable at all. Secondly, it allows us to compare the guarantee stemming from the tail bound to the actual statistical error encountered upon measuring the state. After all, if we were to employ the tail bound on problem instances where E were unknown, we want to first get an initial guess as to how large this overhead may be. Because of these two reasons, we opted to keep the empirical benchmark (despite being numerically inefficient) in the manuscript.

Lastly, we have calculated the uncertainties of the estimators $\Delta\hat{E}$ as suggested by the referee for the various methods over 100 independent runs for the NH₃-molecule:

The Derandomization method lies well off the chart. In this metric, it appears that the Overlapped Grouping measurement delivers the most certain estimates for a large measurement budget. However, the empirical benchmark in our revised script (Figure 3) clearly shows that this method fails to reach comparable accuracy levels of either AEQuO or ShadowGrouping. This corroborates our view that a small variance of the estimator alone is not the defining feature of an accurate measurement routine. To conclude that point, the RMSE is a standard measure of quality to evaluate **the actual error** of an estimator and it is also a standard approach to calculate it in a regime where simulations are possible.

(2) With the proper estimation error provided in our work, bounding the estimated energy as in Eqs. (4) and (5) is seemingly redundant. Even from a practical algorithmic perspective, with our method in Ref. [3] it is possible to keep track of the error contribution of each single Pauli and each group of Paulis and allocate the remaining measurement budget to minimize the error the best. Notice that being able to calculate the proper error of estimation also makes the “truncation criterion” [Eqs. (6) and (7)] redundant, as one eliminates the requirement of trading off “between the statistical error in Eq. (4) and the systematic one [...] introduced [by the truncation criterion]”. In fact, I find that “remov[ing] certain observables from the Hamiltonian” is not well motivated and one of the reasons for which the results in Table I are biased (see below).

Concerning the truncation criterion, our point is that we can mathematically analyse prominent (but heuristic) truncation methods and give an statistically-motivated threshold on when to truncate. For instance and in contrast to popular belief, this threshold is not dependent on the magnitude of the coefficient of the Pauli string. Moreover, in applications outside of quantum chemistry where the accuracy of the measurement can be much less demanding, the truncation criterion can still constitute a valid tool to achieve the same level of accuracy with a smaller total number of measurement rounds. In such scenarios, introducing a systematical estimation error seems practically justified. We commend on the bias in our numerical experiment at a later point below. For the sake of a more streamlined main text, we have shifted the truncation criterion to the Supplementary Information. We agree with the referee’s assessment that it does not yield any contribution for our quantum chemistry benchmark where high precision is paramount.

(3) One of the main results of this work, according to the Authors, is that ShadowGrouping is the first work that “comes with rigorous guarantees and sample complexity bounds for the energy estimation”. While this may be true for shadow methods, it is not true in general. In our work we derive upper bounds on the error (which is tight) and its scaling with respect to the number M of Paulis. This can be seen in the last paragraph of Sec. 3 in Ref. [3], where we also provide equations for these bounds.

We thank the referee to point us to the respective equation. Indeed, it closely resembles our main finding in our work. However, the difference lies in the fine-print: the work of Ref. [3] manages to upper-bound the uncertainty $\Delta\hat{E}$ of any estimator \hat{E} . However, bounding $\Delta\hat{E}$ is not the same as bounding the actual error $|\hat{E} - E|$. Asymptotically, both bounds convey the same. However, at finite measurement rounds, outliers influence the empirical estimator \hat{E} and may substantially move it away from the unknown quantity of interest E . Tail bounds are robust to such cases, yielding confidence intervals unconditionally (i.e. regardless of the measurement outcomes) directly around E with high probability. Our work constitutes a shift away from analysing $\text{Var}[\hat{E}]$ to rather upper-bound $|\hat{E} - E|$. We understand that this fine-print may cause confusion or can easily be overlooked and tried to ensure that it becomes apparent from our improvised version of the manuscript where we give due respect to the findings in Ref. [3].

(4) This work is limited to bitwise commutation. While in the very short term there are hardware limitation for general commutation, the state-of-the-art measurement protocols should allow for choosing whether to employ general commutation or restrict to bitwise. In fact, there are several works proving that general commutation gives a tremendous advantage, particularly for the chemistry examples considered by the Authors.

We thank the referee for this remark. We agree that the presentation of our algorithm for general commutativity relations has been obfuscated. To rectify this, we have majorly extended the explanation how the algorithm works for general commutativity. We agree with the referee’s assessment that general commutativity yield more accurate results, even in the presence of increased noise. Yet, in our numerical benchmark, we have chosen the restriction to bitwise commutativity because of its short-term applicability.

(5) One of the important claims is that ShadowGrouping outperforms other approaches. Looking at Table I, it seems to be the case. However, Table I is obtained with parameters that are (a) not physically relevant and (b) particularly generous to Shadow-based approaches that employ Hamiltonian truncation. Indeed, for $N = 1000$ measurements, several Paulis of the molecules larger than H_2 are never actually measured. Even more problematic, the ShadowGrouping alongside other classical shadow methods (except the Derandomization in [10]) removes from the Hamiltonian-to-be-measured several Pauli terms (see the “truncation criterion” at point 2). As it turns out, the ground states of chemistry molecules are highly biased in the sense that, when all these Paulis are neglected, they significantly improve the resulting estimator. The situation changes dramatically when other input states are considered. For instance, if the Authors were using the ground state of the neglected Pauli as input, they would find out that the advantage of the ShadowGrouping is

severely suppressed, if not fully eliminated. For further information, please see the last two paragraphs of Appendix E in Ref. [3]. My point here is that, from one side, I am not convinced that truncating the Hamiltonian (thus modifying the observable to be estimated) is well motivated. And even if it is, I suspect that the advantage of the ShadowGrouping and the OverlappedGrouping (which precedes this work) methods solely comes from there. From the other side, testing the measurement algorithms for very small numbers of measurements is not physically meaningful. Physical settings most likely require elevated precisions. For instance, as the Authors say, “chemical accuracy is reached below a value of 1.6 mHa”. This requires values of N that are much higher than the $N = 1000$ employed for Table I. Therefore, for Table I to be indicative of an advantage of the ShadowGrouping, N should be large enough to reach chemical accuracy for each molecule.

We thank the referee for his invaluable insights regarding the introduced bias of the methods by neglecting certain cliques. We have expanded our numerical benchmark and (partially) double-checked these assessments. Our findings indicate that the error bound derived from the Derandomization paper does leave many cliques unmeasured, even for $N \gg 10^3$. This is, however, not the case for the error bound derived in our work: here, ShadowGrouping eventually selects all cliques to improve the measurement accuracy over the full range of 10^5 measurement shots. In particular, it does outperform the other state-of-the-art algorithms, including AEQuO for the NH_3 case.

Regarding the truncation, we admit that our presentation, in light of the points of the critique, may be misleading. In the case of quantum chemistry applications, the required accuracy level (chemical accuracy) is too high to make any relevant cuts to the Hamiltonian decomposition. Nevertheless, we want to point out its usefulness once this constraint is less severe: in such situations, the truncation can reduce the number of measurement rounds required to reach the same level of accuracy. We also refer to remark (2) above.

(6) The settings in which AEQuO is run are never explicitly reported. In particular, I am fairly certain that AEQuO’s adaptive features are off, as in Table I AEQuO performs similarly to the Derand [10] protocol (except for the LiH molecule, see next), while in reality adaptive AEQuO consistently outperforms the Derand, even when restricted to bitwise commutation relations (see Appendix E in Ref. [3]). Since the adaptive features of AEQuO come for free (also from a computational point of view), the Authors should include adaptive AEQuO to their results. Secondly, I am puzzled by the fact that non-adaptive AEQuO performs worse than the Derand for LiH, despite from our results the two are completely compatible within statistical error. Finally, the value 7730 for the NH_3 molecule measured with Derand seems off.

For AEQuO, we resorted to the same parameters as reported in Fig. 5 in Appendix E of Ref. [3], i.e. $L = 3$ and $l = 4$ with a total measurement budget of $N = 10^5$ shots. We have now detailed this in the main text of the manuscript as well. As rightfully pointed out, the Derandomization bound solely focusses on the most important clique(s) and thus disregards many others completely. This behaviour is completely tuned by the algorithm’s hyperparameter ϵ . Depending on its choice, the estimation procedure can completely fail as only the dominating clique is measured at all. In our experiments, we resorted to the value of $\epsilon^2 = 0.9$ as reported in Ref. [1]. Unfortunately,

we cannot compare this to the results reported in Ref. [3] as the hyperparameters used for Derandomization have neither been reported. In either case, this does not change any of our conclusion drawn.

(7) What are the errors reported in Table I? Are they – as described in the caption – the “numbers in parentheses” that are apparently missing? If yes, what is the “error [...] in terms of the last two relevant digits”? Is it associated to the expected statistical fluctuation of the RMSE in Eq. (9) from the $N_{\text{runs}} = 100$ repetitions?

We excuse the mismatch between the values in the table and its caption. The RMSE has been calculated 100 independent times and its statistical fluctuation has been captured by calculating the square-root of the empirical standard deviation of $(\hat{E} - E)^2$. However, as per the critique from point 5 above, we have decided against a presentation of the results via a table in the revised version. Here, the uncertainty on the RMSE is included as error bars in the plots. We have expanded the main text to avoid any confusion in the revised manuscript as well.

We would like to thank Luca Dellantonio again for his critical review which has challenged us to improve our numerical demonstration and has helped us to improve the presentation of our manuscript.

In this reply, we have rebutted all raised potential issues and are convinced that our manuscript would now make a valuable contribution in Nature Communications.

References

- [1] Hsin-Yuan Huang, Richard Kueng, and John Preskill. Efficient estimation of Pauli observables by derandomization. *Phys. Rev. Lett.*, 127:030503, Jul 2021.
- [2] Jaouad Mourtada. Exact minimax risk for linear least squares, and the lower tail of sample covariance matrices. *The Annals of Statistics*, 50(4):2157 – 2178, 2022.
- [3] Ariel Shlosberg, Andrew J. Jena, Priyanka Mukhopadhyay, Jan F. Haase, Felix Leditzky, and Luca Dellantonio. Adaptive estimation of quantum observables. *Quantum*, 7:906, 2023.

Reply to Referee 2

We thank the referee for the thoughtful and critical review. We have included versions of our manuscript and Supplementary Information in our resubmission where all changes are highlighted. Now, we go through the major specific points raised by the referee.

As pointed out by the authors, the worst-case bounds are not particularly practical. During a typical VQE run, many states, of varying variances, will be seen. It's certainly true that the worst case bounds are unlikely to be seen, but understanding a more practical example is necessary to demonstrate ShadowGrouping's impact. The authors provide some evidence that ShadowGrouping does perform numerically well in the Table I: Empirical Benchmarks. However, none of the results presented in Table I, ShadowGrouping or otherwise, approach chemical accuracy. They are in fact one or two orders of magnitude away for the fixed shot budget of $N = 1000$. Given the goal of 1.6mHa accuracy, and the results shown in Figure 5 for worst-case measurements required for chemical accuracy, I think that the authors should perform additional numerics to find the 'real-world' number of shots required for chemical accuracy. A numerical study showing the energy error as a function of number of shots, going from the 1000 to whatever is required for chemical accuracy (10^6 ? 10^{10} ?) would greatly strengthen the paper. This would answer a lot of lingering questions about ShadowGrouping.

We thank the referee for pointing out the importance of a numerical investigation in the relevant regime of high accuracy. Indeed, the goal of our quantum chemistry benchmark is to reach a sufficiently high accuracy level (coined chemical accuracy) for many non-commuting Hamiltonian terms. Essentially, we have redone our numerical simulations taking this feedback into account.

We perform a benchmark against state-of-the-art methods up to 10^5 many measurements for various molecules. Across all problem instance sizes considered, only ShadowGrouping and AEQuO yield reliable measurement strategies to actually achieve chemical accuracy, which we show in the **new Figure 3** for the NH_3 example (on 16 qubits). Next, for the generated data, we are able to extract the empirical accuracy as a function of the number of measurement shots. In this way, we extrapolate the number of shots required to reach chemical accuracy, as requested by the referee. Indeed, we observe in our **new Figure 4** that also in this regime of high accuracy ShadowGrouping is more accurate than other methods. In particular, for the NH_3 example, ShadowGrouping requires fewer than 10^7 shots and is roughly 10 times more accurate than AEQuO, which is the next best method from the literature.

Is it consistently better than other methods at different numbers of shots?

Yes, at least for the investigated molecules, see Figure 4. To achieve this advantage for low shot numbers, we have to use our truncation criterion (now in the Supplementary Information IV). However, similar truncations are also performed (implicitly) in the other methods that perform well on low shot numbers.

How much better is ShadowGrouping at convergence? I think the data over many numbers of shots is also useful, as there are proposals for using an adaptive number

| *of shots during optimization runs (see Ref. [1] or Ref. [2] for examples).*

Similar to Figure 4 in the previous version (where we showed the provable accuracy as a function of measurement rounds N), we now track the empirical accuracy as captured by the root mean squared error (RMSE). For better visibility of the respective plots, however, we opted to shift the comparison of guarantees to the Supplementary Information (Supplementary Fig. 2). This allows a direct comparison of both the overhead between the two figures of merit as well as their overall trend.

Moreover, we have increased the number of measurements shots substantially until a clear trend in the empirical data set in. Afterwards, we have used extrapolations to retrieve the actual number of shots required to reach chemical accuracy. Across all problem instance sizes considered, only ShadowGrouping and AEQuO yield reliable measurement strategies to actually achieve chemical accuracy. For smaller problem instances, the additional (co)variance information AEQuO incorporates can be beneficial to further decrease the measurement burden. However, especially for the largest problem instance considered (NH_3), focussing on measuring the most observables simultaneously appears to provide the bigger substantial improvement. This is of particular importance once problem instances beyond the reach of feasible classical methods are considered.

We thank the referees for their suggestion of related literature concerning a possible (adaptive) training procedure of a variational quantum algorithm (VQA) with respect to the required accuracy at each step. At this point, we also thank them for the many smaller typos caught.

| *Is the real-world number of shots significantly lower than the worst case bounds?*

We observe that the worst-case bounds are about two orders of magnitude worse than the ‘real-world’ numbers from our simulations for low to intermediate shot numbers, see our Figure 3. Notably, when fitting the bound by a power law, we find a smaller power law coefficient (around $c \approx 1/3$) compared to the empirical data ($c \approx 0.5$), shown in the upper-right panel. As a consequence, the respective extrapolation to chemical accuracy yields an overhead of more than eight orders of magnitude compared to the empirical data. Nevertheless, our guarantee is the best known so far. We attribute this discrepancy to the empirical error to the fact that our tail bound does not take into account (co)variances between the jointly measured Pauli observables. For instance, in the Methods section IV.E, we show that our tail bound matches the empirical bound in terms of the fit exponent c if and only if all observables commute with each other which is, of course, not the case for the considered quantum chemistry Hamiltonians.

| *However, the provable bound shows an unpractical number of required measurements and the numerical results do not do enough to convince the reader the ShadowGrouping is optimal. For the broad readership of Nature Communications, I believe that there needs to be significantly more clearly defined demonstration that ShadowGrouping is optimal. As it stands now, the results are of interest to a smaller community. As such, I cannot accept the paper unless major revisions can more conclusively demonstrate ShadowGrouping’s benefits.*

Inspired by the referee’s feedback we have repeated our numerical analysis in a practically relevant setup and prepared a major revision of our manuscript. Due to our Proposition 4, obtaining optimal measurement schemes is a computationally hard task and, hence, optimality is a subtle issue. In principle, one can brute-force the calculation

of an optimal measurement scheme but only for very small examples.

However, as explained above, we have now performed a more clearly defined demonstration that ShadowGrouping is certainly the most competitive method for the energy estimation with Pauli basis measurements and, additionally, it is supported by rigorous guarantees.

We would like to thank again the referee for their constructive criticism and hope that the new numerical demonstration renders our manuscript suitable for publication in Nature Communications.

References

- [1] Jonas M. Kübler, Andrew Arrasmith, Lukasz Cincio, and Patrick J. Coles. An adaptive optimizer for measurement-frugal variational algorithms. *Quantum*, 4:263, May 2020.
- [2] Matt Menickelly, Yunsoo Ha, and Matthew Otten. Latency considerations for stochastic optimizers in variational quantum algorithms. *Quantum*, 7:949, March 2023.

Reply to Referee 1

We thank Luca Dellantonio for his continued critical review effort and address his criticism in the following. We have highlighted the major text changes in **green** in the revised manuscript.

(1) After reading the Authors' response, my opinion is unchanged. I will not address point-by-point the Authors' response, as my review would look very similar to the first one.

We would have been very interested to know the specifics of what parts of our reply do not appeal to you. In our revision, we have made serious efforts to adequately motivate our different stance towards the core problem in a point-by-point fashion and why we disagree on critical items. Using the same approach, we address the remaining points of concern below.

(2) The disagreement mostly stem from fundamental assumptions/axioms, such as whether one should employ the ACTUAL error, which is unaccessible, or an ESTIMATED error, which can be accessed.

We explained the choice of error measure in our last reply and it is really unfortunate that the reviewer does not seem to have the time to exactly say at what point he disagrees with our arguments.

In short, we think that the actual error should be provided as long as it is accessible. This is particularly important when the estimated error comes without rigorous gaurantees, such as tail bounds. For instance, the error estimates from Ref. [1] come without such rigorous guarantees. As such, one may easily spoof the error estimator, especially in cases where the total measurement budget is severely limited. In such a case, the estimator effectively becomes a biased one while its (empirical) variance may appear very small, possibly leading to large actual error. Even for moderate measurement budgets, the sheer amount of Pauli terms to be measured typically leads to a set of measurement settings of very skewed relative frequency. Again, the low frequency results in a meager absolute number of shots which, in turn, may by chance result in a low recorded empirical variance even if the actual variance is significantly larger.

Ironically, the ongoing discussions on our numerically results concerning the actual error tremendously helped us with debugging and fine-tuning the algorithms from the literature (namely Derand and OGM, see also remark (3) below) for a fair comparison.

(3) I do believe that the comparison with other approaches is not very precise. I am not speaking only of AEQuO, whose performance is still surprisingly poor. As can be seen in Fig. 3 of the revised manuscript, the Derand and Overlapped Grouping (OG) methods reach horizontal plateaus. In the case of the Derand, I believe that this is due a rounding error in the code, that occurs for large numbers of measurements. In this case, the Derand code starts allocating measurements to the XXX...XXX clique only, which would explain the horizontal plateau. In the case of the OG protocol, I believe that the authors do not appropriately set a very hidden parameter inside the OG code, that determines how many and which Paulis are to be disregarded in the estimation process. This explains the plateau since, when enough many measurements are allocated, these Pauli starts contributing to a shift away the exact average.

These two (plausible) imprecisions are certainly not the Authors' fault, yet still make their benchmark not very accurate.

We thank the referee to point us to the technical fine-prints of either method and address them in detail in the following. Indeed, the numerical implementation of the Overlapped Grouping (OG) method requires a hyperparameter T , related to the total measurement budget N_{tot} , which has been preset to $T = 1000$. This effectively leads to a truncation of the larger Hamiltonians, hence the missing convergence of the method, i.e., the observed plateau. In our updated benchmark, we now report the OG results for a hyperparameter value $T = 10^7 > N_{\text{tot}}$ which resolves the plateaus. However, this update does not alter any of our conclusions drawn previously in terms of the observed overall accuracy of the method. We thank Luca Dellantonio for rightfully identifying

this inappropriately tuned hyperparameter value of the previous version.

The derandomization method (Derand) is very perceptible to the choice of the hyperparameter ϵ . A too small value simply results in solely picking the Pauli term of largest magnitude and leaving most terms unmeasured (yielding the observed plateaus in our benchmark). On the other hand, a value too large eradicates this bias at the expense of weak overall performance (most terms are still measured seldomly and the focus on terms of largest magnitude is dwindling). While a full analysis of Derand lies well outside the scope of our manuscript, we contrast these insights to ShadowGrouping. The latter does not require any hyperparameter fine-tuning at this stage (we commend on this in the Methods Section IV.C around eq. (25)) while leaving no observable unmeasured by construction. We try to quantify the respective performances in terms of $\Sigma = \sum_{i=1}^M N_{\text{comp}}^{(i)} |h_i|$ in the bar plot below. Here, $N_{\text{comp}}^{(i)}$ denotes how many compatible measurement settings have been found for each of the Pauli terms of magnitude $|h_i|$. The intuition behind this choice is that a suitable measurement scheme ought to measure the terms with large magnitude more often than less important ones while trying to measure as many terms jointly as possible. This is a computationally efficient measure (no actual measurement have to be performed for this comparison) at the expense, of course, of being slightly distorted by neglecting any (co)variance information. For Derand, we subsume the respective values by $\max_{\epsilon} \Sigma(\epsilon)$, $\epsilon \in \{10^{-4}, \dots, 10^0\}$. Since ShadowGrouping outperforms Derand in all observed settings (in the particular case of BeH_2 where it does not, the respective Derand method does not measure around 84% of the terms; excluding the corresponding ϵ puts Derand behind ShadowGrouping in terms of Σ again), we have refrained from a further empirical performance analysis of Derand in our benchmark and have commented on these conclusion in the main text.

We would like to thank Luca Dellantonio again for his critical review which keeps challenging us to improve our numerical demonstration.

In this reply, we have rebutted all raised potential issues and are convinced that our manuscript would now make a valuable contribution in Nature Communications.

References

- [1] Ariel Shlosberg, Andrew J. Jena, Priyanka Mukhopadhyay, Jan F. Haase, Felix Leditzky, and Luca Dellantonio. Adaptive estimation of quantum observables. *Quantum*, 7:906, 2023.

Reply to Referee 2

We thank the referee for the continued thoughtful and critical review. We have highlighted the major text changes in green in the revised manuscript. Now, we go through the major specific points raised by the referee.

The authors have performed many additional numerical experiments to address my comments from the previous review. Namely, they performed additional numerical tests up until 10^5 shots and then extrapolated to get the number of shots required for chemical accuracy. This certainly increases confidence in this work and demonstrates that the ShadowGrouping method should generally outperform the other methods.

We again thank the referee for having challenged us to improve our numerical benchmark and for his assessment of our previous revision.

However, I wonder why the authors decided to extrapolate, rather than directly perform the experiments, up until 10^7 or 10^8 shots. It is certainly nice to have the exponent c , but it would be nice to see ShadowGrouping actually achieve chemical accuracy for the largest system. While I understand that this will be at least $100\times$ more costly than the experiments already performed, this gives the appearance of a bottleneck somewhere. Systems that are much larger than these are the ones that we really care about, where we would be running directly on a quantum computer with, say, $100+$ qubits. A really, really rough extrapolation from H_2 (8 qubits, around 2×10^5 expected shots) to NH_3 (16 qubits, around 5×10^6 shots) would say the number of shots goes up by more than an order of magnitude when the number of qubits doubles. Will ShadowGrouping be able to perform at 10^9 shots?

We thank the referee for this thoughtful suggestion and have increased the total number of actual samples collected to verify that ShadowGrouping reliably reaches chemical accuracy as previously claimed. We have done so by taking the previously recorded value for the required total number of measurement rounds, increased it roughly by a factor of two and generated the corresponding number of measurement settings with ShadowGrouping. The measurement procedure has been repeated a hundred times independently to gather statistics again. We report the average absolute error $|\hat{E} - E|$ between the ground-state energy E and ShadowGrouping's estimate \hat{E} . Lastly, we normalize this value by the chemical accuracy $\epsilon_{\text{acc.}}^{\text{chem.}} = 1.6$ mHa for better clarity in the plot. This result constitutes our **New Figure 4b** in the Results section (and reprinted below for the referee's convenience). In all reported cases of the numerical benchmark, we reliably reach this target accuracy (colored boxes indicate the standard deviation with the black dot indicating the mean). To compare to the extrapolation of the empirical data, we show the respective accuracy predictions (including the uncertainties from the fitting procedure) as gray boxes. They agree arguably well with the extrapolation, deeming ShadowGrouping fit to provide a suitable measurement strategy even for large numbers of measurement rounds.

We thank the referee to point us to this previous apparent weakness in the demonstration of

our algorithm. The limiting factor has been the simulation of the actual measurement process as it involves the state simulation. To visualize this directly, we also report the average wall time to either generate a single measurement setting via ShadowGrouping or to measure the state in the corresponding basis below. We see that the measurement procedure constitutes the practical bottleneck in our benchmark due to its exponential scaling (line added to guide the eye). In contrast, generating measurement settings (which anyways have to be done only once) roughly follows the $M \log M$ -scaling where $M = \mathcal{O}(n^4)$. We do agree that for large systems, the overall wall time may still contribute a limiting factor from a practical perspective. In such cases, however, we note that writing out the Pauli decomposition with $\mathcal{O}(M)$ will be challenging already. We therefore believe ShadowGrouping suitable for all practically feasible cases, even for larger systems in the future.

I think, with minor revisions, this manuscript could be accepted into Nature Communications. Whether this is through an additional simulation explicitly showing ShadowGrouping achieving chemical accuracy, or a discussion about scaling towards larger number of shots, I leave up to the authors.

We would like to thank again the referee for their continued helpful feedback and hope the improved numerical demonstration where chemical accuracy is reliably reached for all considered problem instances answers them affirmatively.

Reply to Referee 3

We thank the referee for the thoughtful and critical review.

We have highlighted the major text changes in green in the revised manuscript. Now, we go through the specific points raised by the referee.

This manuscript, "Guaranteed Efficient Energy Estimation of Quantum Many-Body Hamiltonians using ShadowGrouping," proposes a novel strategy for estimating the expectation value of a Hamiltonian. The numerical results demonstrate that their method is state-of-the-art. The manuscript is clearly written and provides a detailed comparison of their algorithm with existing ones. In my opinion, it might be suitable for the Nature Communications Journal. However, I have some concerns.

We thank the referee for his assessment of our manuscript. We hope that the points of critique rightfully raised below are sufficiently and affirmatively answered.

(1) I could not find useful information about your algorithm from Figure 1. I suggest the authors describe how to generate Q_1, \dots, Q_N in the figure, which should be the main contribution of this paper.

We thank the referee for pointing to this issue. Figure 1 has been placed both as an overview of our main results (i.e., the ShadowGrouping algorithm and the tail-bound guarantee) as well as an emphasis on the fact that the energy estimation task concerns both the preprocessing as well as the postprocessing of the actual measurement. As such, we believe that the technical fine-print of the algorithm itself is not suited to be incorporated into the figure at this point of the manuscript.

In order to provide more useful information about our algorithm, we have improved the consistency of its caption and added a forward-reference to the respective subsection in our manuscript and thank the referee for bringing this to our attention.

(2) As suggested by other reviewers, it seems the author did not run AEQuO's code correctly, such as by not turning on the adaptive features. The authors should carefully check if they really made a mistake, and give an explicit explanation in the manuscript.

We have thoroughly checked the code for errors and have cross-compared it to its source [1]. Indeed, the adaptive features are turned on.

In order to provide a more immediate demonstration of the adaptive feature (and its limitation), we consider the following toy Hamiltonian with its corresponding qubit-wise commutativity graph:

$$H = Z\mathbf{1} + \mathbf{1}Z + X\mathbf{1} + \mathbf{1}X \quad \begin{array}{cc} Z\mathbf{1} & - & \mathbf{1}Z \\ | & & | \\ \mathbf{1}X & - & X\mathbf{1} \end{array}$$

This graph allows for four cliques with respective measurement bases ZZ, ZX, XZ, XX . The AEQuO algorithm starts out by employing the largest-degree-first (LDF) algorithm to find a clique covering of this Hamiltonian once. Without loss of generality, we assume it picks the disjoint cliques $\{Z\mathbf{1}, \mathbf{1}Z\}$ and $\{X\mathbf{1}, \mathbf{1}X\}$ which can be measured in the Pauli- ZZ and the XX -basis, respectively. Afterwards, the adaptive feature of AEQuO guides how many measurement shots are allocated for each of these two cliques, based on previous measurement outcomes. This way, the state dependence enters into the algorithm. To visualize the limits of this adaptability, we consider all two-qubit tensor product combinations of the eigenstate $|0\rangle$ of the Pauli- Z -operator and $|+\rangle$ of the Pauli- X -operator. We run AEQuO for each of these 4 states and investigate how often either clique is to be measured. The algorithm is set up such that it allocates 50 settings non-adaptively at first (i.e., alternating between either of the two cliques) and updates the empirical covariance estimate afterwards. Then, another 50 settings are allocated

adaptively, effectively taking information about the respective state into account. We summarize the respective frequencies in the figure below. Here, the two leftmost (colored in blue) bars correspond to the first non-adaptive allocation round and the other two (orange-colored) bars to the adaptive allocation round afterwards. Initially, both settings have been selected non-adaptively equally often. However, the adaptive capability of AEQuO ensures to measure cliques with larger empirical uncertainty more frequently. Measuring the state $|00\rangle$ in the ZZ -Pauli basis (i.e., the first clique) always yields the outcome $[1, 1]$ but a random outcome in the XX -Pauli basis (i.e., the second clique). As a result, the first clique is seldomly selected to be measured after taking the previous measurement data into account (upper-left panel). The roles of ZZ and XX are reversed when considering the $|++\rangle$ -state into account (lower-right panel). If neither measurement setting yields a smaller variance (as for the states $|0+\rangle$ and $|+0\rangle$), the adaptive feature of AEQuO does not improve the measurement routine at all. In order to capture such corner cases, the clique-finding algorithm (such as LDF) has to take into account the measurement outcomes as well. This is not taken into account by AEQuO. In all fairness, the current version of ShadowGrouping may also occasionally run into the same problem when trying to brake ties in the corresponding weight function (as it would randomly select to measure among the ZZ, XZ, ZX, XX -bases. We, however, meticulously explain this toy example to illustrate that the AEQuO-algorithm does not adaptively adjust the clique covering (but only their respective repetitions) of the commutativity graph. In contrast, ShadowGrouping does not impose a fixed clique covering but can implicitly pick from the set of all the suitable cliques by virtue of our tail bound (while never explicitly constructing it). Given the quite inhomogenous distribution of the coefficients' magnitudes in the Hamiltonian decomposition, we strongly believe that this yields a more drastic improvement compared to the adaptive adjustment of measurement repetitions of a fixed (non-optimal) clique covering. To further corroborate this insight, we find that AEQuO only improves over ShadowGrouping for the H_2 -molecule (we refer to the **reworked Figure 4** in our main text for the comparison) where the clique cover returned by the LDF algorithm is essentially optimal. In all other cases, and especially for the largest system sizes, ShadowGrouping drastically improves upon AEQuO.

For NH_3 , we have collected further evidence on why this is the case. To this end, we inspect the generated measurement settings for both AEQuO (in the adaptive setting as indicated in the main text) and ShadowGrouping. We find that ShadowGrouping employs a drastically larger number of distinct measurement settings (relating to distinct cliques in the underlying commutativity graph) which, additionally, also make better usage of each single shot: for the latter, we count how

many qubits remain unmeasured and display the respective histograms below (the numbers in parantheses indicate the number of distinct settings). Given that the NH_3 -molecule is represented by 16 qubits, we find that AEQuO does not make use of a significant portion of its collected measurement data in terms of not measuring certain qubits in a measurement round. As indicated in the outlook of our main manuscript, we leave the combination of the two advances as a future research avenue.

Finally, we have opted to exclude this detailed analysis of AEQuO out of our revised manuscript as to not shift the focus away from ShadowGrouping itself. As compromise, we have inserted a short summary sentence to provide a brief explanation for its succesful performance in the manuscript.

(3) *The other reviewer proposes the following comment, and I also agree with them. “first and foremost, we provide rigorous sampling complexity upper bounds for the read-out of quantum experiments as needed for the direct energy estimation of quantum-many body Hamiltonians.” This statement is too strong, many other measurement schemes also have similar upper bounds.*

We thank the referee for this remark concerning the corresponding part of the discussion in our manuscript. We agree that our previous statement has been too strong a statement. In our revised script, we have toned-down our statement slightly to reflect the referees’ remarks.

(4) *Prove that the method does not rely on the specific initial state; demonstrate that it remains efficient for any state, not just the ground state.*

We thank the referee for this suggestion. Indeed, ShadowGrouping does not depend on the choice of the quantum state to be measured: it only takes into account the Hamiltonian, i.e., its decomposition in terms of Pauli operators. The generation of the next measurement basis is inspired by our energy estimation tailbound (theorem 1 in the main manuscript) which, in turn, holds uniformly for any quantum state. Therefore, we expect that the energy accuracy reached with ShadowGrouping at given shot number N has a scaling independently of N for any quantum state. To test this numerically, we have extended our benchmark accordingly. As shown in our **New Figure 5** in the Methods section and as an extended version below for the referee’s convenience, we prepare various other quantum states and measure their energies with respect to the same Hamiltonian for the H_2 molecule encoded in the 6-31g basis set, comprising eight qubits. Since ShadowGrouping does not depend on the quantum state to be measured, we generate up to 10^5 measurement settings in total only once and measure the respective quantum states with these settings. As states, we have chosen to measure thermal states at a given inverse temperature β and Haar-random states, including global depolarization noise with parameter p . We plot the energy profiles as function of their corresponding parameters in the left column, respectively. We pick around ten different choices of parameters, prepare the corresponding quantum state and measure it in the generated measurement bases and track the

average empirical deviation over 100 independent runs (and 100 different random Haar states) between the energy estimate \hat{E} of ShadowGrouping and the correct energy E for the respective quantum state. This procedure is repeated for each fermion-to-qubit mapping chosen in the main text. All states exhibit a similar accuracy level in the order of 10^{-2} mHa and, moreover, the same scaling with respect to N up to a multiplicative factor which is attributed to the state-dependent variance of the energy estimator. Importantly, even measuring the arguably most challenging state, the maximally-mixed state (orange lines) only adds a constant factor of at most three to the observed accuracy level. Since our numerical benchmark now includes both pure states (both structured as the ground state and unstructured as Haar random ones) and mixed states (including two different interpolations to the maximally mixed one), we believe it a compelling demonstration of ShadowGrouping’s applicability to the energy estimation task for arbitrary quantum states.

(5) What do the values in Table II of the manuscript represent? Are they variance or energy? I couldn’t find any description clarifying this.

We thank this referee for pointing this out. They are the root-mean squared error of the estimates to the actual ground-state energy in mHa. We have corrected the corresponding caption in our revised manuscript.

We would like to thank the referee again for his critical review which has challenged us to improve our numerical demonstration and has helped us to improve the presentation of our results. We hope that we have convinced the referee that our manuscript would now make a valuable contribution in Nature Communications.

References

- [1] Ariel Shlosberg, Andrew J. Jena, Priyanka Mukhopadhyay, Jan F. Haase, Felix Leditzky, and Luca Dellantonio. Adaptive estimation of quantum observables. *Quantum*, 7:906, 2023.

Dear Editor,

Before writing my report, I would like to specify that I am an author of another approach for measuring quantum observables. Namely, AEQuO, in Ref. [6]. In the decision of not recommending this work for publication, I have tried and hopefully succeeded in being objective. As I am most familiar with AEQuO, in the following I will refer to it on several occasions for comparison.

In their work, the Authors propose an approach for estimating quantum observables. Despite the paper is well written, I believe that the ShadowGrouping technique is missing properties that a measurement approach should have. In all fairness, most of the following points are not addressed by any of the classical shadow measurement protocols (e.g., OverlappedGrouping, Derand, Adaptive Pauli). Therefore, this review can be considered more generally as a critique towards this family of methods. The editor could perhaps keep this in mind when considering the following points. I would like to remark that I consider classical shadow techniques extremely powerful in other settings.

1. The performance of measurement protocols cannot be tested with the estimator in Eq. (9), as the exact value of the quantity being measured is generally unknown. For a measurement protocol to be meaningful, aside from the estimated average value it must provide a meaningful estimate for the error (not a bound). In our work, we provide the theory to estimate the proper error (see Eqs. (2) in <https://arxiv.org/pdf/2110.15339.pdf>), that can be adapted to any measurement protocol – including all ShadowGrouping variants. In Table I and for all numerical results, the authors should employ an estimator that is agnostic of the exact value of the observable to be measured. As a side remark, our error estimator in Eqs. (2) from <https://arxiv.org/pdf/2110.15339.pdf> can be employed to give the statistically exact estimation error without the requirement of repeated iterations that are numerically expensive¹. This can be very helpful when presenting the numerical results, since statistical fluctuations from repeated measurements can be large. Also, it can come at handy for my criticism number 5 below.
2. With the proper estimation error provided in our work, bounding the estimated energy as in Eqs. (4) and (5) is seemingly redundant. Even from a practical algorithmic perspective, with our method in <https://arxiv.org/pdf/2110.15339.pdf> it is possible to keep track of the error contribution of each single Pauli and each group of Paulis² and allocate the remaining measurement budget to minimize the error the best. Notice that being able to calculate the proper error of estimation also makes the “truncation criterion” [Eqs. (6) and (7)] redundant, as one eliminates the requirement of trading off “between the statistical error in Eq. (4) and the systematic one [...] introduced [by the truncation criterion]”. In fact, I find that “remov[ing] certain observables from the Hamiltonian” is not well motivated and one of the reasons for which the results in Table I are biased (see below).
3. One of the main results of this work, according to the Authors, is that ShadowGrouping is the first work that “comes with rigorous guarantees and sample complexity bounds for the energy estimation”. While this may be true for shadow methods, it is not true in general. In our work we derive upper bounds on the error (which is *tight*) and its scaling with respect to the number M of Paulis. This can be seen in the last paragraph of Sec. 3 in <https://arxiv.org/pdf/2110.15339.pdf>, where we also provide equations for these bounds.
4. This work is limited to bitwise commutation. While in the very short term there are hardware limitation for general commutation, the state-of-the-art measurement protocols should allow for

¹ This is done by employing the exact variances and covariances of the Paulis in the Hamiltonian in the formula of Eqs. (2) in our work. Notice that this can be done because one can obtain these quantities directly from the experiment (see <https://arxiv.org/pdf/2110.15339.pdf>).

² Even when the groups change in the measurement process, which allowed AEQuO to be adaptive.

choosing whether to employ general commutation or restrict to bitwise. In fact, there are several works proving that general commutation gives a tremendous advantage, particularly for the chemistry examples considered by the Authors.

5. One of the important claims is that ShadowGrouping outperforms other approaches. Looking at Table I, it seems to be the case. However, Table I is obtained with parameters that are (a) not physically relevant and (b) particularly generous to Shadow-based approaches that employ Hamiltonian truncation³. Indeed, for $N = 1000$ measurements, several Paulis of the molecules larger than H_2 are never actually measured. Even more problematic, the ShadowGrouping alongside other classical shadow methods (except the Derandomization in [10]) removes from the Hamiltonian-to-be-measured several Pauli terms (see the “truncation criterion” at point 2). As it turns out, the ground states of chemistry molecules are highly biased⁴ in the sense that, when all these Paulis are neglected, they significantly improve the resulting estimator⁵. The situation changes dramatically when other input states are considered. For instance, if the Authors were using the ground state of the neglected Pauli as input, they would find out that the advantage of the ShadowGrouping is severely suppressed, if not fully eliminated. For further information, please see the last two paragraphs of Appendix E in <https://arxiv.org/pdf/2110.15339.pdf>.

My point here is that, from one side, I am not convinced that truncating the Hamiltonian (thus modifying the observable to be estimated) is well motivated. And even if it is, I suspect that the advantage of the ShadowGrouping and the OverlappedGrouping (which precedes this work) methods solely comes from there. From the other side, testing the measurement algorithms for very small numbers of measurements is not physically meaningful. Physical settings most likely require elevated precisions. For instance, as the Authors say, “chemical accuracy is reached below a value of 1.6 mHa”. This requires values of N that are much higher than the $N = 1000$ employed for Table I. Therefore, for Table I to be indicative of an advantage of the ShadowGrouping, N should be large enough to reach chemical accuracy for each molecule.

6. The settings in which AEQuO is run are never explicitly reported. In particular, I am fairly certain that AEQuO’s adaptive features are off, as in Table I AEQuO performs similarly to the Derand [10] protocol (except for the LiH molecule, see next), while in reality adaptive AEQuO consistently outperforms the Derand, even when restricted to bitwise commutation relations (see Appendix E in <https://arxiv.org/pdf/2110.15339.pdf>). Since the adaptive features of AEQuO come for free (also from a computational point of view), the Authors should include adaptive AEQuO to their results. Secondly, I am puzzled by the fact that non-adaptive AEQuO performs worse than the

³ I am not implying, in any way, that the Authors intentionally chose parameters that favour their approach. In fact, the same parameters are generally employed for benchmarking all measurement protocols (including AEQuO). However, in our case, this bias was against AEQuO and we demonstrated that, despite it, our protocol is often advantageous compared to others. Here, the Authors concluded that their approach is advantageous without considering that the advantage very likely comes from this bias only.

⁴ As can be seen in Fig. 5 of <https://arxiv.org/pdf/2110.15339.pdf>, both the Adaptive Pauli Shadow [11] and Overlapped grouping strategy are characterized by smaller errors (compared to AEQuO) for small budgets. We have conducted massive amounts of testing to understand why, and eventually concluded that it comes entirely from neglecting part of the observable-to-be-measured. By running AEQuO on the truncated observables, we were able to greatly and consistently outperform both Overlapped Grouping and Adaptive Pauli Shadow for all measurement budgets. However, we also realized that the advantage was highly dependant on the input state. For carefully chosen ones (or random ones) this advantage completely disappeared.

⁵ Notice that this also explains why ShadowGrouping and the OverlappedGrouping (that are the methods that most prominently employ this Hamiltonian truncation) perform so well when the Hamiltonian get larger while the measurement budget is unchanged.

Derand for LiH, despite from our results the two are completely compatible within statistical error. Finally, the value 7730 for the NH₃ molecule measured with Derand seems off.

7. What are the errors reported in Table I? Are they – as described in the caption – the “numbers in parentheses” that are apparently missing? If yes, what is the “error [...] in terms of the last two relevant digits”? Is it associated to the expected statistical fluctuation of the RMSE in Eq. (9) from the $N_{\text{runs}} = 100$ repetitions?

These are the main reasons for which I cannot recommend this work for publication in Nature Communications. I thank again the Editor for the opportunity to provide feedback on this work, and truly hope that my criticisms were taken as constructive as possible.

My best regards,

Luca Dellantonio

Main results: This manuscript, "Guaranteed Efficient Energy Estimation of Quantum Many-Body Hamiltonians using ShadowGrouping," proposes a novel strategy for estimating the expectation value of a Hamiltonian. The numerical results demonstrate that their method is state-of-the-art. The manuscript is clearly written and provides a detailed comparison of their algorithm with existing ones. In my opinion, it might be suitable for the Nature Communications Journal. However, I have some concerns:

(1) I could not find useful information about your algorithm from Figure 1. I suggest the authors describe how to generate Q_1, \dots, Q_N in the figure, which should be the main contribution of this paper.

(2) As suggested by other reviewers, it seems the author did not run AEQuO's code correctly, such as by not turning on the adaptive features. The authors should carefully check if they really made a mistake, and give an explicit explanation in the manuscript.

(3) The other reviewer proposes the following comment, and I also agree with them.

"first and foremost, we provide rigorous sampling complexity upper bounds for the read-out of quantum experiments as needed for the direct energy estimation of quantum-many body Hamiltonians."

This statement is too strong, many other measurement schemes also have similar upper bounds.

(4) Prove that the method does not rely on the specific initial state; demonstrate that it remains efficient for any state, not just the ground state.

(5) What do the values in Table II of the manuscript represent? Are they variance or energy? I couldn't find any description clarifying this.